# Anthropogenic stressors impact fish sensory development and survival via thyroid disruption

Marc Besson [1,2,3 ✉], William E. Feeney [4,5,6], Isadora Moniz[1], Loïc François [1], Rohan M. Brooker [7], Guillaume Holzer [8], Marc Metian [3], Natacha Roux[1,2], Vincent Laudet [2,10,11] & David Lecchini[1,9,11]

Larval metamorphosis and recruitment represent critical life-history transitions for most teleost fishes. While the detrimental effects of anthropogenic stressors on the behavior and survival of recruiting fishes are well-documented, the physiological mechanisms that underpin these patterns remain unclear. Here, we use pharmacological treatments to highlight the role that thyroid hormones (TH) play in sensory development and determining anti-predator responses in metamorphosing convict surgeonfish, *Acanthurus triostegus*. We then show that high doses of a physical stressor (increased temperature of +3 °C) and a chemical stressor (the pesticide chlorpyrifos at 30 μg L$^{-1}$) induced similar defects by decreasing fish TH levels and affecting their sensory development. Stressor-exposed fish experienced higher predation; however, their ability to avoid predation improved when they received supplemental TH. Our results highlight that two different anthropogenic stressors can affect critical developmental and ecological transitions via the same physiological pathway. This finding provides a unifying mechanism to explain past results and underlines the profound threat anthropogenic stressors pose to fish communities.

---

[1] PSL Research University: EPHE-UPVD-CNRS, USR 3278 CRIOBE BP 1013, 98729 Papetoai, Moorea, French Polynesia. [2] Observatoire Océanologique de Banyuls-sur-Mer, UMR CNRS 7232 BIOM, Sorbonne Université, 1 Avenue Pierre Fabre, 66650 Banyuls-sur-Mer, France. [3] International Atomic Energy Agency - Environment Laboratories, 4a Quai Antoine 1er, Principality of Monaco 98000, Monaco. [4] Environmental Futures Research Institute, Griffith University, Nathan, Australia. [5] Department of Behavioural Ecology and Evolutionary Genetics, Max Planck Institute for Ornithology, Seewiesen, Germany. [6] Department of Zoology, University of Cambridge, Cambridge, UK. [7] Centre for Integrative Ecology, School of Life and Environmental Sciences, Deakin University, Geelong, VIC, Australia. [8] Institut de Génomique Fonctionnelle de Lyon, ENS Lyon, Lyon, France. [9] Laboratoire d'Excellence "CORAIL", Moorea, French Polynesia. [10] Present address: Marine Eco-Evo-Devo unit, Okinawa Institute for Science and Technology (OIST), 1919-1 Tancha, Onna-son, Okinawa 904-0495, Japan. [11] These authors contributed equally: Vincent Laudet, David Lecchini. ✉email: marc.besson@ens-lyon.org

The variety, frequency, and amplitude of anthropogenic stressors faced by organisms is unprecedented and increasing[1,2]. The scale of human activities has led to once localized perturbations, such as heat waves[3] and waterborne chemical toxicants[4] becoming issues of global importance. Exposure to stressors can directly impact how organisms function and also decrease their resilience to further exposure or exposure to other stressors, resulting in cascading negative repercussions for them and the ecosystem they inhabit[5,6]. The negative impacts of anthropogenic stressors can be felt by all organisms at all life stages; however, aquatic species may be more vulnerable than terrestrial organisms[7]. Further, disruptions at critical life-history transitions, such as during metamorphosis[8], can disproportionately threaten the structure of communities and persistence of species[9].

Teleost fishes represent almost half of the world's vertebrates and generally exhibit a metamorphic transition from their larval to juvenile phase[10,11]. In marine species, such as coral and temperate reef fishes, this transition typically occurs as larvae move from the pelagic environment to their juvenile/adult reef-associated habitat, i.e., recruitment[11]. The sensory systems of many recruiting fishes undergo rapid development as they metamorphose, which may help them locate and navigate toward suitable settlement habitats[12,13], and survive during this life-history transition[14,15]. Nevertheless, fishes often experience an extreme predation bottleneck during recruitment; for instance, recruiting coral-reef fishes have an estimated average mortality rate of 30% per day as a result of predation[16]. Myriad recent studies have reported that various anthropogenic stressors, such as artificial light pollution[17], increasing temperature[18–20], ocean acidification[21,22], sound pollution[23,24], and waterborne chemical pollutants[11,14,25] can lead to the deterioration of ecologically important behaviors, including the ability to detect predators and avoid predation. Predators and prey do not appear to be equally affected by anthropogenic stressors[21], suggesting that alterations to predator–prey dynamics have the potential to drastically reshape fish communities[26–28]. Despite this, our understanding of the mechanism(s) that underpin the patterns reported in these studies remains limited[29,30].

The endocrine system plays an important role in post-embryonic developmental processes[8,31]. In teleost fishes, metamorphosis is controlled by thyroid hormones (TH), including thyroxine ($T_4$) and triiodothyronine ($T_3$)[10]. Environmental stressors (e.g., decreasing water level and food deprivation) and anthropogenic stressors (e.g., waterborne chemical pollutants) have the ability to disrupt thyroid function, which can affect the timing, duration, and completion of metamorphic processes[32,33]. Surprisingly, the ultimate consequences of stressor-induced developmental and endocrine impairments, such as their impact on predator–prey dynamics, remain poorly studied[34]. TH plays a key role in neurogenesis and sensory development in vertebrates[35,36], and are central for regulating recruitment in reef fishes[11]. Thus, TH signaling disruption may present a generalizable physiological mechanism through which reports of behavioral and survival deteriorations in recruiting fishes following exposure to anthropogenic stressors can be understood[34].

Here, we use the coral-reef-dwelling convict surgeonfish, *Acanthurus triostegus*, to investigate the importance of the TH endocrine pathway on sensory development during metamorphosis, whether exposure to two distinct anthropogenic stressors (increased temperature and the waterborne organophosphate pesticide chlorpyrifos) can cause TH signaling disruption, and whether the resulting impacts on sensory development increase vulnerability to predation.

## Results

### TH signaling controls the development of sensory structures.
To study sensory development during metamorphosis, wild settlement-stage *A. triostegus* (day 0 = d0, Fig. 1a) were collected at night, while they were moving from the pelagic environment to the reef, using a crest net[11]. Following capture, they were maintained for up to 8 days (d8) in in situ cages, during which they fully completed metamorphosis[11]. The olfactory, visual, and mechanosensory structures (Fig. 1b–d) showed rapid development from d0 to d8 in control individuals, with the development of new lamellae, a 50% increase in bipolar cell density in the retina, and a 240% surge in trunk canal pore density, respectively (Fig. 1e–g). We investigated the role of TH in the development of these sensory structures by injecting fish daily from d0 to d5 with different pharmacological drugs: (i) $T_3$ + iopanoic acid (T3 treatment) to achieve TH signal activation, and (ii) NH3 (a TH antagonist, N3 treatment) to achieve TH signal disruption[11]. The T3 treatment induced an accelerated development of all three sensory structures, with elevated densities of bipolar cells at d2, as well as elevated number of lamellae and greater density trunk canal pores at both d2 and d5 (Fig. 1e–g). In contrast, N3-treated fish experienced repressed sensory development, at both d2 and d5 for the retina and the olfactory organ (Fig. 1e, f), and at d2 for the trunk canal (Fig. 1g). Supplemental $T_3$ rescued olfactory organ development in N3-treated fish (Supplementary Fig. 1).

### TH signaling modulates behavioral responses to predator stimuli and vulnerability to predation.
As predation is high in the 2 days following settlement in reef fishes[16], we examined the behavioral outcomes of different sensory structure developmental states by testing whether d0 and d2 control, T3-, and N3-treated fish responded differently when presented with chemical and visual cues from a common reef predator: the blacktail snapper, *Lutjanus fulvus*. In chemical choice experiments, d2 control fish showed a clear avoidance of predator cues (Fig. 2a). This response was similar in T3-treated fish, while N3-treated fish did not discriminate between water sources, similar to d0 fish (Fig. 2a). When comparing responses to visual cues in a separated aquarium, we again found that d2 control and T3-treated fish avoided predator cues, while N3-treated fish showed no preference, similar to d0 fish (Fig. 2b). To investigate whether this inability to identify predation risk makes them more vulnerable to predation, we conducted a predation test in an in situ arena. We found that d2 T3-treated fish exhibited significantly higher survival (i.e., number of fish that survived) than d2 control fish, while d2 N3-treated fish exhibited lower survival than d2 control fish (Fig. 2c).

### Increased temperature and chlorpyrifos affect TH levels and sensory development.
To examine whether independent exposure to two anthropogenic stressors can affect TH levels and sensory system development, we exposed *A. triostegus* from d0 to d2 to increased temperature and to the organophosphate pesticide chlorpyrifos (CPF). Fish were exposed to either: (i) water at 28.5 °C (control), 30.0 °C ( + 1.5 °C), or 31.5 °C ( + 3.0 °C), following end-of-century projections for the tropical Pacific[37]; or (ii) water containing no CPF (control), acetone (solvent control, CPF0), or CPF at 1, 5, and 30 µg L$^{-1}$ (CPF1, CPF5, and CPF30, respectively)[11]. We found that fish exposed to +3 °C experienced a significant reduction in $T_4$ and $T_3$ levels at d2 compared with control fish (32% and 35%, respectively), while no difference was observed in fish exposed to +1.5 °C compared with control fish (Fig. 3a, b). Fish exposed to +3 °C also exhibited a significant reduction in both bipolar cell and trunk canal pore density (7 and 9%), but exhibited no difference in the number of lamellae compared with control fish (Fig. 3c–f). We observed a significant reduction (25%) in $T_4$ levels at d2 in CPF30 fish compared to control fish, while there were no differences between control, CPF0, CPF1, and CPF5 fish (Fig. 3a). We also found no

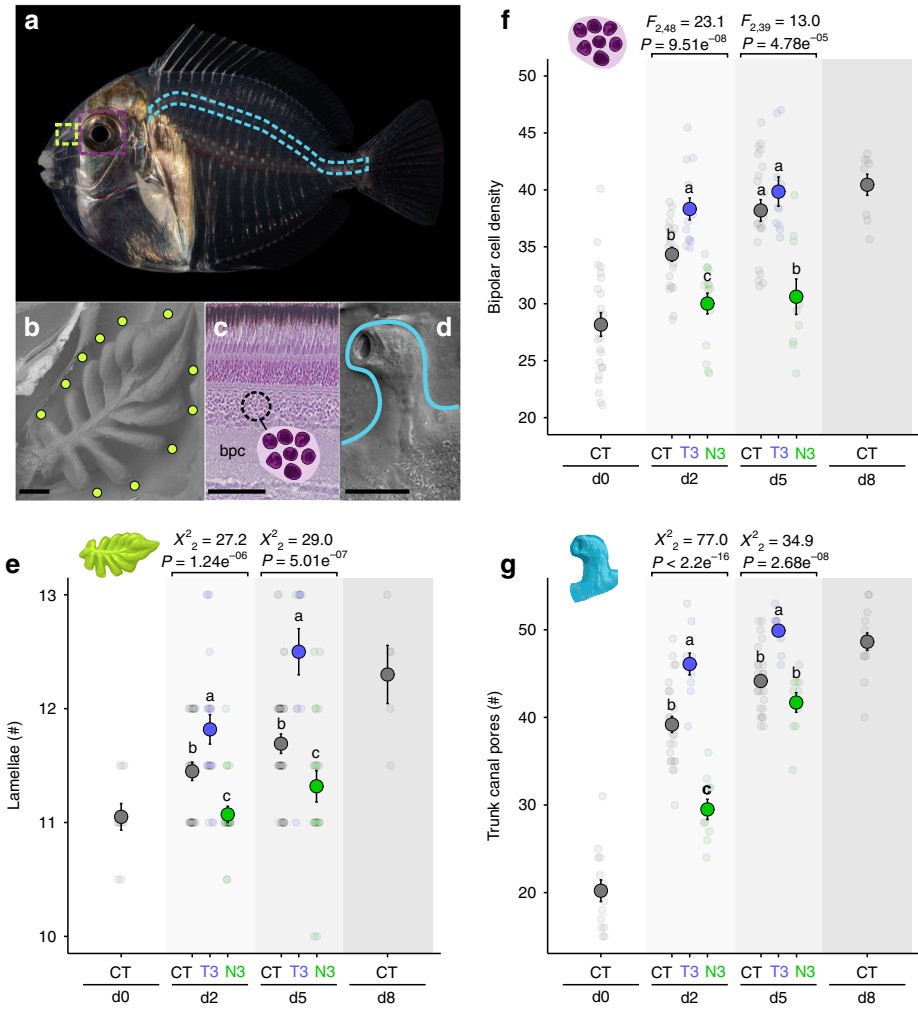

**Fig. 1 Development of sensory structures in *Acanthurus triostegus* at recruitment. a** Metamorphosing larva of *A. triostegus* (standard length = 2.5 cm). Left-to-right dotted-areas indicate the regions where b, c and d, are found, respectively. **b** Scanning electron microscope (SEM) photograph of an olfactory organ rosetta, with 11 lamellae (green dots). Scale bar: 100 μm. **c** Cross-section in the retina presenting the bipolar cells (bpc) region. Scale bar: 50 μm. **d** SEM photograph of a trunk canal pore (demarcated by a blue line). Several trunk canal pores are found in the blue dotted area in the panel **a** (Supplementary Fig. 4). Scale bar: 100 μm. **e–g** Changes in lamellae number ($n = 162$), bipolar cell density ($n = 125$), and number of trunk canal pores ($n = 120$), respectively, along recruitment (from day 0 (d0) to d8) and across hormonal treatments (CT = control; T3 = T3 treatment; N3 = N3 treatment). Data are indicated as mean (opaque circles) ± SE (error bars), and transparent circles indicate each data point. Letters indicate statistically different groups according to two-sided Tukey post hoc tests following COM-Poisson GLM ($\chi^2$) or LM (*F*). Source data are provided as a Source Data file.

differences in $T_3$ levels between control, CPF0, CPF1, and CPF5 fish, but significantly reduced $T_3$ levels in CPF30 fish (28%) compared with control fish (Fig. 3b). CPF30 fish also exhibited reduced densities of bipolar cell (10%) and trunk canal pores (15%), but no difference in the number of lamellae compared with the control fish (Fig. 3c–f). Exposures to +3 °C or CPF30 were therefore associated with comparably affected TH levels and development of sensory structures, but CPF30 fish experienced a lower T3/T4 ratio than +3.0 °C fish (Supplementary Fig. 2).

**T₃ injections can restore survival in stressor-exposed fish**. To investigate whether increased susceptibility to predation was causally related to anthropogenic stressor-induced TH disruption, we conducted a predation test. We compared the survival of d2 fish that were exposed to: (i) control conditions, +3 °C, or +3 °C with supplemental $T_3$; and (ii) control conditions, CPF30, or CPF30 with supplemental $T_3$. When compared with control fish, those exposed to +3 °C and CPF30 experienced lower survival (Fig. 3f). Interestingly, both +3 °C and CPF30 fish treated with

supplementary $T_3$ experienced higher survival compared with untreated +3 °C and CPF30 fish, respectively (Fig. 3f).

**Increased temperatures and chlorpyrifos synergistically decrease TH levels**. To test whether exposure to both stressors had additive or interacting effects, we simultaneously exposed d0 larvae to different suboptimal dose regimes of both stressors (i.e., dose regimes that were found to individually have no significant effect on TH levels) until d2. We found that co-exposure to +1.5 °C and CPF1 did not significantly reduce $T_4$ or $T_3$ levels, but that co-exposure to +1.5 °C and CPF5 significantly decreased $T_4$ and $T_3$ levels (43% and 26%, respectively) (Fig. 4a, b).

**Discussion**
Our results highlight the key role an endocrine process (i.e., TH signaling) plays in regulating sensory system development in recruiting fishes, the effect anthropogenic stressors can have on this endocrine function, and the consequences of decreased TH levels and affected sensory development for determining the

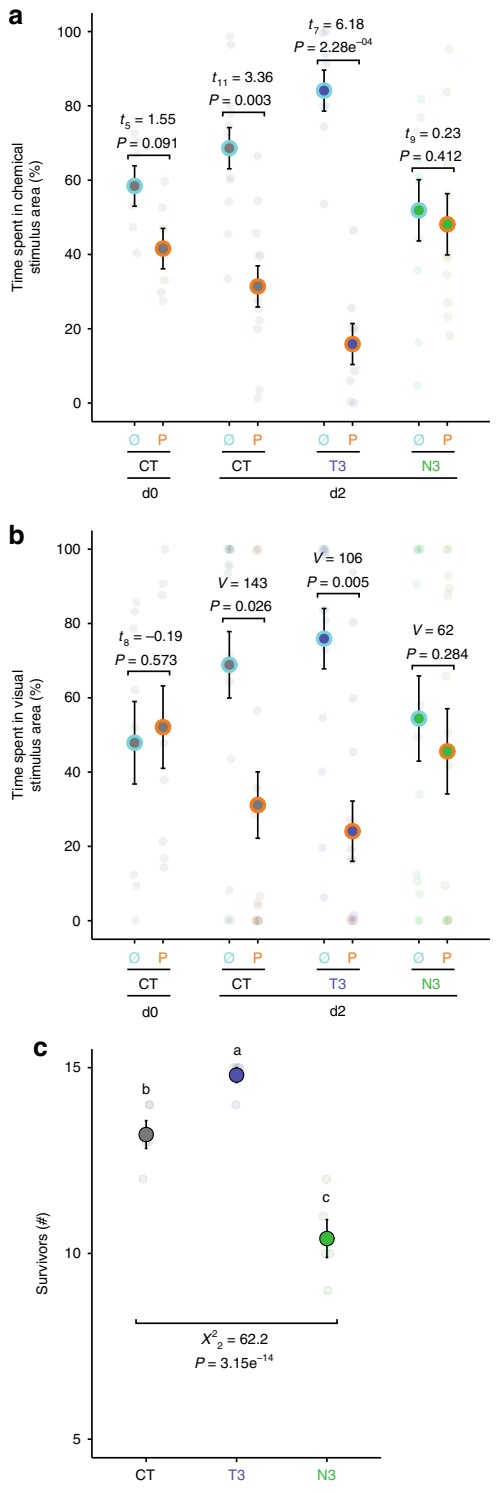

**Fig. 2 Thyroid hormone signaling and *Acanthurus triostegus* sensory abilities and survival facing predation. a** Chemical preference when facing seawater containing no-predator cue (Ø) vs seawater containing predator cue (P) along recruitment (from day 0 (d0) to d2) and across hormonal treatments (CT = control; T3 = T3 treatment; N3 = N3 treatment); $n = 72$. **b** Visual preferences when facing visual cues from two separate aquaria: a control "fish-empty" tank (Ø) vs tank with a live predator (P), across hormonal treatments; $n = 114$. Statistical results in **a** and **b** indicate the outputs from one-sided paired $t$ tests or Wilcoxon signed-rank tests. **c** Differential survival in a predation arena of d2 *A. triostegus* across hormonal treatments; $n = 225$ fish in $n = 5$ replicates. Data are indicated as mean (opaque circles) ± SE (error bars), and transparent circles indicate each data point. Letters indicate statistically different groups according to two-sided Tukey post hoc tests following COM-Poisson GLM ($\chi^2$). Source data are provided as a Source Data file.

in teleost fishes[10,11], our results indicate that TH disruption may provide a unifying physiological mechanism that can help explain reports of deteriorated anti-predator behaviors following exposure to an array of anthropogenic stressors[21,23,25].

Thyroid hormones are important for regulating sensory development in teleost fishes[11,14,36,38], and our results provide insights into the role they play in this development during recruitment. *A. triostegus* exhibits an enclosed nose (Supplementary Fig. 3a), well-formed rosette lamellae (Supplementary Fig. 3b), and a fully formed trunk canal (Supplementary Fig. 4a) at the time of settlement, but relatively low cell densities in the retina (Fig. 1f; Supplementary Figs. 6–8)[13,39]. This suggests that, with the possible exception of the retina, the sensory structures of *A. triostegus* are generally more developed than other reef[40–42] and non-reef[36,43] fish species at a similar life-history stage. Despite this advanced stage of development, we found that fish with pharmacologically enhanced TH signaling experienced faster sensory development, those with disrupted TH signaling (either pharmacologically (Fig. 1e–g) or environmentally (Fig. 3c, e)) experienced diminished sensory development, and those with pharmacologically disrupted TH that received supplemental $T_3$ experienced rescued development of their olfactory organ (Supplementary Fig. 1). This development of new lamellae promoted by TH is consistent with neurogenesis[35], as the number of lamellae roughly correlates with the number of olfactory neurons[44]. In contrast, the development of the trunk canal involves both the differentiation of canal neuromasts (i.e., neurogenesis, not studied here) and the enclosure and elaboration of the canal itself[43]. The epithelial proliferation that occurs, that is relevant here, is the formation and elongation of the pores in the skin covering the lateral line canal. The fact that lamellae development was not affected by increased temperature or CPF (Fig. 3d) suggests that the endocrine disruption caused by these stressors may not be severe enough (e.g., in comparison with the N3 treatment) to affect this developmental process. These results highlight the sensitivity and complexity of the mechanisms underlying the actions of TH on sensory system development, offering an avenue for future research, in particular on the auditory system, as hearing was not studied here but is also used by larval fish during recruitment and impacted by stressors[23].

In addition to causing developmental defects, TH signaling disruption also affected the behavior and survivorship of metamorphosing fish. Fish that experienced pharmacological TH disruption presented anti-predator behaviors that were comparable with pre-metamorphosed larvae and suffered increased mortality during a predation experiment (Fig. 2a–c). In contrast, fish that received supplemental $T_3$ experienced a marked increase in their capacity to avoid predation (Figs. 2c and 3f). We found

outcome of predator–prey interactions. Fish with pharmacologically disrupted TH signaling experienced diminished sensory development and anti-predator behaviors, and were more vulnerable to predation. Sensory development and vulnerability to predation were comparably affected by two distinct anthropogenic stressors (increased temperature and CPF). However, fish that received supplemental $T_3$ experienced an apparently recovered ability to avoid predation, implicating TH signaling disruption as a mechanism associated with decreased survival following exposure to these stressors. In addition to the recognized role of TH for successful metamorphosis and recruitment

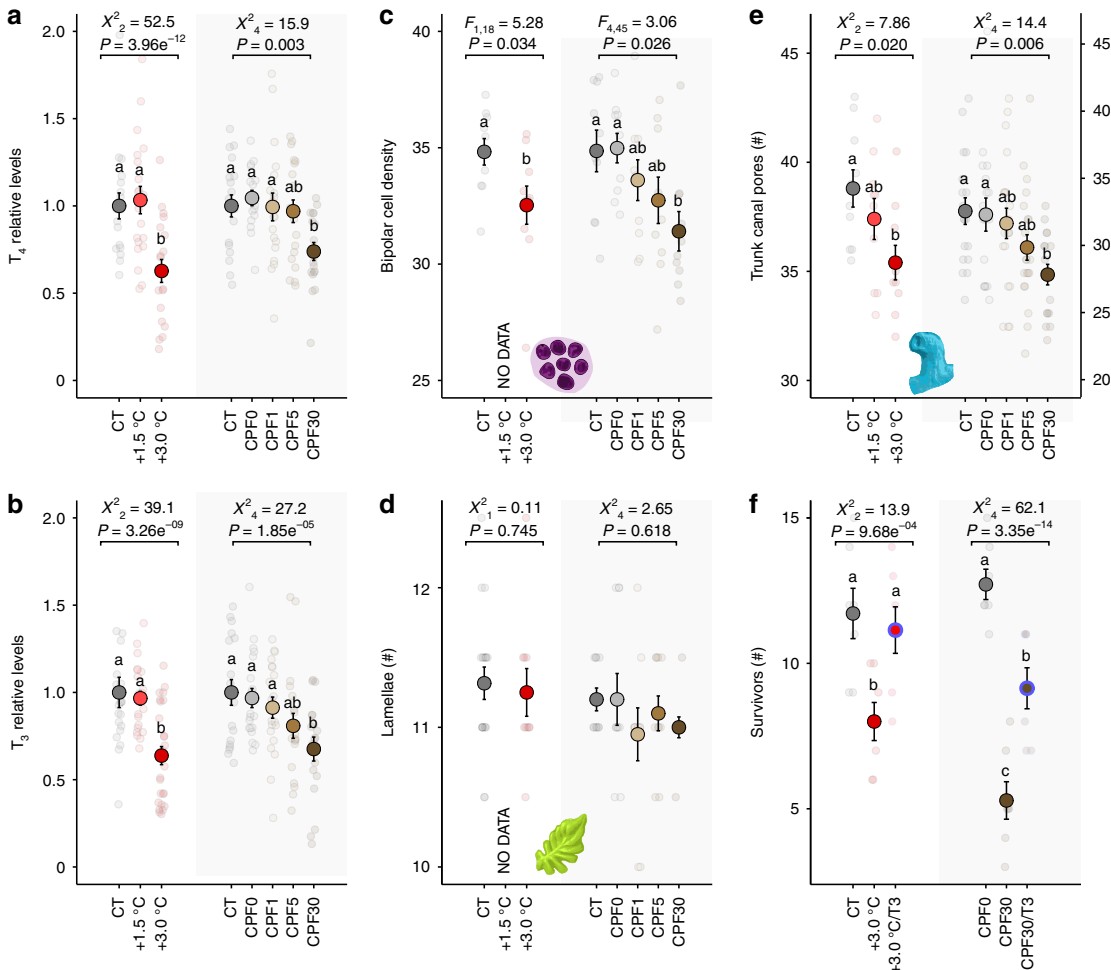

**Fig. 3 Effects of increased temperature and chlorpyrifos on TH levels, sensory organ development, and survival. a–e** Respective $T_4$-levels ($n = 150$), $T_3$-levels ($n = 167$), bipolar cell densities ($n = 70$), number of lamellae ($n = 79$), and lateral line canal pores ($n = 123$) in d2 *A. triostegus* exposed to increased temperature (CT = control, 28.5 °C; + 1.5 °C = 30.0 °C; + 3.0 °C = 31.5 °C; white panels) and CPF (CT = control, no CPF; CPF0 = acetone solvent control; CPF1, CPF5 and CPF30 = CPF at 1, 5, and 30 µg $L^{-1}$, respectively; gray panels). **f** Differential survival, in a predation arena, of d2 *A. triostegus* exposed to increased temperature or CPF and across hormonal treatments ($n = 630$ fishes in $n = 7$ replicates). Data are indicated as mean (opaque circles) ± SE (error bars), and transparent circles indicate each data point. Letters indicate statistically different groups according to two-sided Tukey post hoc tests following Gamma GLMEM ($\chi^2$ in **a**, **b**), LM (*F*), or COM-Poisson GLM ($\chi^2$ in **d**–**f**). Source data are provided as a Source Data file.

that T3-treated fish exhibited fastened sensory development (Fig. 1e, f) and comparable responses to predator cues to control fish (Fig. 2a, b), which is consistent with an accelerated sensory development facilitating more effective anti-predator abilities. Nevertheless, we cannot rule out the possibility that supplemental $T_3$ may have also impacted their behavior in other ways. For example, supplemental $T_3$ may affect swimming behavior (e.g., as suspected in *Danio rerio*, McMenamin pers. comm.) and might alter the cues emitted by these individuals (e.g., smelling or tasting different to the predators), which may have contributed to their ability to avoid predation (Fig. 2c).

While exposure to high levels of increased temperature and CPF both inhibit sensory development and reduce a fish's likelihood of avoiding predation, the two stressors affected endocrine signaling differently. Temperature had a higher impact on $T_4$ levels than CPF (Fig. 3a), as indicated by a higher $T_3/T_4$ ratio (Supplementary Fig. 2). This suggests a greater central effect, most probably at the neuroendocrine level, which may alter thyroid activity and thus explain the decreased levels of $T_3$ that we observed[45]. In contrast, CPF had a greater effect on $T_3$ levels than temperature (Fig. 3b), as indicated by a lower $T_3/T_4$ ratio (Supplementary Fig. 2), suggesting a more downstream or

peripheral effect, possibly on $T_3$ metabolism[46,47]. This variety of action modes may explain the synergistic effect of both stressors on TH levels with co-exposure to 1.5 °C or CPF5 decreasing TH levels (Fig. 4a, b), while these stressor levels did not affect TH levels when exposed separately (Fig. 3a, b). Further, as the actions of TH are temperature-specific[48], the toxicity of TH-disrupting pollutants such as CPF may intensify with increasing temperatures. Given that stressors are rarely experienced individually, these results highlight the vulnerability of aquatic organism endocrine functions and the risk posed by anthropogenic stressors[7,9].

Reef fish-recruitment areas (i.e., nurseries) are typically coastal, shallow, and often tidally isolated, making them highly sensitive to dramatic environmental changes, such as sudden heat waves[49,50] and terrestrial pollutant runoff[51], that are expected to increase in both frequency and severity globally. Organophosphates are the most widely used pesticide globally, and CPF is the second most commonly detected pesticide in water, frequently occurring in temperate and tropical coastal runoff[52]. Under these circumstances, acute fluctuations such as temperature spikes of +1.5 °C and +3.0 °C are environmentally relevant, and CPF levels from 1 to 30 µg $L^{-1}$ are informative in the context

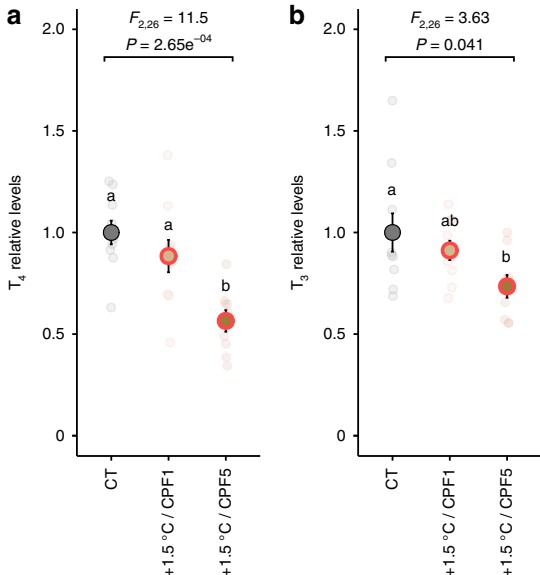

**Fig. 4 Effects of increased temperature and chlorpyrifos co-exposure on TH levels. a, b** Respective $T_4$ levels ($n = 29$) and $T_3$ levels ($n = 29$) in d2 *A. triostegus* exposed to increased temperature (CT = control, 28.5 °C; + 1.5 °C = 30.0 °C) and CPF (CT = control, no CPF; CPF1 and CPF5 = CPF at 1 and 5 µg L$^{-1}$, respectively). Data are indicated as mean (opaque circles) ± SE (error bars), and transparent circles indicate each data point. Letters indicate statistically different groups according to two-sided Tukey post hoc tests following LM (*F*). Source data are provided as a Source Data file.

of decreasing water quality in coastal areas in response to increasing pesticide use and land clearing[51]. Indeed, rapid temperature shifts are common in coastal surface waters and can reach up to 12 °C following solar and tidal forcing[50]. Likewise, while CPF levels in contaminated surface waters are generally below 1 µg L$^{-1}$ with limited persistence in the water column, these levels can spike up to 26 µg L$^{-1}$ in rivers[53]. This suggests that following run-offs, and on a short timescale such as the 32-h exposure of our study, CPF levels in coastal waters could largely exceed the pg/ng per liter concentrations usually reported in seawater[54–56]. Acute exposure to increased temperatures of +1.5 and +3 °C or CPF levels spanning from 1 to 30 µg L$^{-1}$ therefore reflects the temperature fluctuations[49,50] and potential or future pesticide fluctuations[53] that larval fishes may experience when recruiting to coastal nurseries under high anthropogenic influence. Larger and older fishes are less vulnerable to TH signaling disruption and associated neurological defects than metamorphosing fishes[57], indicating that exposure to acute stressors may alter fish predator–prey dynamics during this critical temporal window. As small declines in survival during recruitment can have dramatic consequences for population replenishment, our results raise concerns about the future of coastal fish nurseries, which, under the threats of climate change and anthropogenic stressors, could turn into ecological traps[58].

Overall, this study highlights that short-term exposure to acute anthropogenic stressors has adverse consequences for the development, behavior, and survival of metamorphosing fish by affecting their TH endocrine function. Given the widespread nature of metamorphosis in aquatic organisms[8] and that metamorphosis is typically regulated by sensitive endocrine pathways[59], this study underlines the profound threat posed by anthropogenic stressors to the function, resilience, and persistence of aquatic populations and communities.

## Methods

**Ethics statement**. This study did not involve endangered or protected species and was carried out in accordance with the guidelines of the French Polynesia code for animal ethics and scientific research (https://www.service-public.pf/diren/partager/code/). All protocols and experiments were approved by the CRIOBE-IRCP animal ethics committee (DL-20150214).

**Model species**. The convict surgeonfish *Acanthurus triostegus* (Linnaeus, 1758) is an abundant coral-reef-associated species found throughout the Indo-Pacific[60], including around Moorea Island, French Polynesia[11]. It has a pelagic larval duration of ~53 days[60] after which larvae move back to the reef[61]. *A. triostegus* is a well-studied species with regards to metamorphosis, with a well-defined developmental sequence[11]. The recruitment of *A. triostegus* larvae to reef habitats coincides with a true metamorphosis into juveniles, with this full process taking around 1 week[11]. Its metamorphosis is controlled by thyroid hormones (TH, the precursor thyroxine ($T_4$), and the active hormone triiodothyronine ($T_3$)), which is the same as other teleosts and other metamorphosing vertebrates such as amphibians[11]. TH signaling in larval *A. triostegus* is vulnerable to disruption by anthropogenic stressors, including the waterborne pesticide chlorpyrifos and artificial light at night[11,17]. Given the importance of TH during teleost metamorphosis[10], and as metamorphosis coincides with recruitment in a taxonomically diverse range of coral-reef fishes (i.e., Acanthuridae, Apogonidae, Balistidae, Chaetodontidae, and Pomacentridae)[11], we consider *A. triostegus* a representative teleost model for examining the effects of anthropogenic stressors on fish recruitment via their impacts on metamorphic processes.

**Study period and site**. This study was conducted from February 2015 to June 2018, at Moorea Island, French Polynesia (S17°32′16.4589″, W149°49′48.3018″). Sampling for the examination of sensory development under pharmacological treatments was conducted in 2015. Sampling for the investigation of behavioral preferences and survival to predation under pharmacological treatments, as well as sensory development under anthropogenic stressor exposure, were conducted in 2016. Sampling for the examination of survival to predation under anthropogenic stressors exposure, and TH levels under co-exposure to anthropogenic stressors were conducted in 2018. Sampling for the investigation of TH levels under single anthropogenic stressors was conducted in both 2016 and 2018.

**Fish sampling**. Settlement-stage *A. triostegus* (i.e., fully transparent individuals[11], here define as day 0 (d0) individuals), were collected on the north–east coast of the island (S17°29′49.7362″, W149°45′13.899″) at night using a crest net[11], as they transitioned from the ocean to the reef. Fish were then transferred to either in situ cages (for thyroid hormone signaling experiments) where they remained until d8, or to aquaria at the CRIOBE Marine Research Station (for increased temperature and chlorpyrifos exposure experiments) where they remained until d5.

**Thyroid hormone signaling experiment: control, T3- and N3 treatments**. To test the role of TH on sensory development, behavior (response to sensory cues), and survival (predation test) in metamorphosing *A. triostegus*, their TH pathway was pharmacologically manipulated. Fish were injected, at d0 immediately following capture, in their ventral cavity with 20 µl of a pharmacological treatment: (i) $T_3$ + iopanoic acid (IOP) both at $10^{-6}$ M (T3 treatment), or (ii) NH3 at $10^{-6}$ M (N3 treatment). IOP was used as an inhibitor of deiodinase enzymes, following comparable work in mammals and amphibians[62], and as routinely used in fish to prevent the immediate degradation of injected $T_3$[48]. The T3 treatment was therefore applied to promote TH signaling. NH3 is a known antagonist of TH receptors (TR) in vertebrates[63] and in *A. triostegus* in particular[11]. NH3 prevents the binding of TH such as $T_3$ to TR, therefore impairing the binding of transcriptional coactivators to TR, which therefore remain in an inactive and repressive conformation[63,64]. The N3 treatment was thus applied to repress TH signaling by disrupting the TH pathway. $T_3$ and NH3 were initially suspended in dimethyl sulfoxide (DMSO), at $10^{-2}$ M, and then diluted in phosphate buffered saline (PBS) 1× to reach $10^{-6}$ M. Control fish were injected with 20 µL of DMSO diluted 10.000 times in PBS 1× to control for the effect of the solvent and injection. DMSO and NH3 non-toxicity has been previously determined[11]. Until subsequent sampling, all fish were re-treated each morning (i.e., at d2, d3, and d4 for fish sampled at d5) to maintain pharmacological activity[11].

**Thyroid hormone signaling experiment: fish husbandry**. Following collection and subsequent treatment, larvae were transferred to a nursery area on the north coast of the island (S17°29′26.5378″, W149°53′29.2252″) where they were raised in in situ cages (cylindrical cages, diameter: 30 cm; height: 50 cm, 15 fish per cage). This allowed them to develop in in situ conditions[11]. As *A. triostegus* feeds on algal turf following settlement[11,60], cages were stocked with a supply of turf-covered coral rubble that was replaced daily, ensuring both shelter and constant food availability. Fish were subsequently sampled on d2, d5, and d8, post collection to examine sensory development, behavior (response to sensory cues), and survival (predation test). Twelve in situ cages were used, and the use of any given cage was randomized prior each experiment.

**Increased temperature and chlorpyrifos (CPF) exposures: fish husbandry and treatments**. Following collection, d0 fish were maintained in groups of ten individuals at the CRIOBE Marine Research Station. Each group was held in a 30 L × 20 W × 20 H-cm aquaria containing 12 L of filtered (1-μm filter) seawater. All tanks were subject to a 12:12 h light–dark cycle (06:00–18:00 light period) and oxygenated with an air stone. Twelve aquaria were used (six for exposures to increased temperatures only, and six for exposures with CPF), and the use of any given aquarium was randomized prior each experiment.

For increased temperature treatments, seawater was in an open system, and water temperature was maintained at either 28.5 °C, 30.0 °C or 31.5 °C. 28.5 °C was chosen as the basal temperature as this was the mean temperature in the Moorea lagoon at the time of the study, and corresponds to the mean annual lagoon temperature in this region (http://observatoire.criobe.pf). Subsequent increases of +1.5 °C and +3.0 °C were selected, as these are in line with end-of-century projections for tropical Pacific sea surface temperatures[37]. Fresh coral rubble was added to the tanks and replaced each day to provide both food and shelter. Heaters controlled by thermostats were used to maintain the temperature treatments. Before the experiment, each tank was in open circuit with temperature maintained at 28.5 °C. At the beginning of the experiment, fish were introduced in the aquarium, and the thermostat temperature was then set up to 28.5 °C, 30.0 °C or 31.5 °C according to the treatment. The temperature of interest was reached within 2 h. Temperature in each tank was then visually checked (on the thermostat controller) at least five times per day, and never differed from the target temperature by more than 0.2 °C.

For CPF exposure, five different treatments were applied: unaltered seawater (control), seawater with acetone at a final concentration of 1:1.000.000 (CPF0, solvent control treatment, as CPF was made soluble using acetone), or seawater with CPF at a nominal concentration of either 1, 5, or 30 μg L⁻¹ (CPF1, CPF5, and CPF30 treatments), based on the findings of recent studies of reef fishes exposed to CPF[11,14,65]. CPF was spiked in each tank from dilutions that were prepared in advance: 1 μg μL⁻¹, 5 μg μL⁻¹, and 30 μg μL⁻¹. From these dilutions, 12 μL were pipetted and spiked in the 12-L exposure tanks, therefore reaching nominal concentrations of 1 μg L⁻¹, 5 μg L⁻¹, and 30 μg L⁻¹. Similarly, 12 μL of acetone was spiked in the tank for the CPF0 condition. Spike was allowed to mix for 2 min (water mixing due to the air stone) before fish were introduced in the tank. At the end of the 32-h exposure, CPF concentrations in the water or in the fish tissues were not evaluated, as we were only interested in the effects of CPF spikes on fish metamorphic processes. Nevertheless, a previous study using similar methods and nominal concentrations of similar magnitude (i.e., ranging from 4 to 64 μg L⁻¹) measured CPF levels corresponding to 80% of nominal concentrations after 24 h[66], therefore suggesting a good stability of CPF levels in the condition of our study. Environmentally, these nominal concentrations represent high (CPF1) to extremely high (CPF30) exposures, as recorded contamination concentration in Australian and North American surface waters are generally below 1 μg L⁻¹, with a few high outliers reaching up to 26 μg L⁻¹[53]. However, this shows that concentrations of up to 30 μg L⁻¹ are possible on a short timescale (such as the 32-h exposure of this study), in particular in coastal and shallow areas such as fish nurseries. Aquaria used for CPF exposure treatments and associated controls were not equipped with coral rubble to prevent potential interaction with the pesticide[11]. This may explain the delay in trunk canal development observed in control fish from the CPF exposure treatments compared with control fish from the increased temperature treatments and control fish from the TH treatments (Figs. 1g and 3e). Water was replaced each day to ensure the maintenance of CPF concentrations[11].

For combined increased temperature and CPF exposure treatments, fish were also maintained without coral rubble with water replaced daily.

For treatments where fish were exposed to an anthropogenic stressor and provided with supplemental T₃, fish were maintained in either the elevated temperature or CPF exposure treatments as described above, but were also injected with either T₃ (T3 treatment) or with DMSO (control, to control for solvent and injection). These injections were done as described above (thyroid hormone signaling experiment: control, T3- and N3-treatments).

**Sample preparation and fish measurements**. For thyroid hormone quantifications and histological analyses, fish were first euthanized in freshly prepared MS222 at 0.4 mg ml⁻¹ in filtered seawater at 4 °C, and instantly placed on ice. Following euthanasia, all fish except those for histological analyses were weighed (W) then placed on a scale bar and photographed for standardized measurements (e.g., height (H) and standard length (SL)). These measurements were used to calculate Fulton's condition factor $K = 100*(W/SL^3)$ (with SL, in cm, and W, in g) for each individual[67]. Weight measurements were performed using a precision balance (*Ohaus Adventurer Precision*), and length measurements were taken using the *imageJ* software.

**Retinas**. The retina is the light-sensitive tissue within the eye, and is composed of different cell layers organized ventrally and dorsally around the optic nerve (Fig. 1c). To analyze the retina's cell layers, we removed the head region of euthanized fish in a way that allowed us to distinguish the right retina from the left retina, and the dorsal and ventral sides of the retina. This region was then fixed in Bouin solution before being embedded in paraffin for microtome sectioning (5 μm). Sections were then stained using hematoxylin and eosin. We chose cross-

sections of the retina that were done at similar depth by selecting sections at the level of the optical nerve (Supplementary Fig. 5a). These sections enabled us to identify different cell segments, types, and layers, among which (i) photoreceptor external segments (perceiving light signals), (ii) photoreceptor nuclei, (iii) bipolar cells (which integrate the synaptic signals originating from the photoreceptors), and (iv) ganglion cells (which integrate signals from bipolar cells and create action potential toward the optic nerve) were easily distinguishable (Supplementary Fig. 5b). We only investigated the right eye of *A. triostegus* fish, as evidence suggests this species is visually lateralized at recruitment, and predominantly uses its right eye to examine visual predator stimuli[14]. Also, we only examined the dorsal side of the retina, as the ventral side (vs) was shown to not undergo change at metamorphosis in another coral-reef fish species, the goatfish *Upeneus tragula*[13] (Supplementary Fig. 5a). To compare the developmental state of the retina between treatments, we looked at the peripheral area of the dorsal side (see the dotted square in Supplementary Fig. 5a, magnified in Supplementary Fig. 5b) as settlement-stage individuals in another acanthurid species (*Naso brevirostris*) showed weak if any spatial specialization of the retina, in particular on this axis[68]. We prioritized the examination of bipolar cell densities as important changes in bipolar cell density was observed during *U. tragula* metamorphosis[13]. However, we also examined the densities of photoreceptor external segments, photoreceptor nuclei, and ganglion cells along metamorphosis and across TH treatments, revealing for photoreceptor external segments (Supplementary Fig. 6) and nuclei (Supplementary Fig. 7) a TH-dependent density increase at metamorphosis, while ganglion cells showed no density variation during metamorphosis (Supplementary Fig. 8). Cell counting was performed in a 50-μm wide area perpendicular to the retina cell layers (Supplementary Fig. 5b). The measure of bipolar cell density corresponds to the number of bipolar cells per 0.001 mm² in the inner-nuclear cell layer (INL), after measuring the mean thickness of this INL (average of three measurements at random non-overlapping locations within the 50-μm wide area) (Supplementary Fig. 5b). The prn and ggc densities were obtained in a similar manner after measuring the mean width of their respective cell layer. Pes density corresponds to the exact number of pes in the 50-μm wide area as the pes cell layer is monocellular (Supplementary Fig. 5b). Cells that were located on the edges of the 50-μm wide area were included in the counts.

**Olfactory organ lamellae**. In fish, the olfactory organ is a fluid-filled blind sac that contains the ciliated sensory epithelium, which forms a rosette comprise of a number of folds, i.e., lamellae[44]. In *A. triostegus*, the left and right olfactory organs can be found in two cavities on the dorsal surface, between the eye and the snout edge, with water moving through each cavity from the anterior nostril to the posterior nostril (Supplementary Fig. 3). Pictures (e.g., Supplementary Fig. 3) were obtained using SEM microscopy following specialized tissue preparation involving the dissection of the olfactory organ (removal of the thin skin layer between the two nostrils) and fixation in a sodium cacodylate + glutaraldehyde solution (2.5% glutaraldehyde in 1 M sucrose and 0.1 M sodium cacodylate, pH 7.4) for a week at 4 °C. Samples were then washed (10% sucrose in 0.1 M sodium cacodylate solution, pH 7.4) 15 × 15 min at room temperature (RT). Samples were then dehydrated using successive ethanol (EtOH) baths (30%, 50%, 70%, 85%, 95%, 100%, 100%, 100%, each time during 15 min at RT). Samples were then placed in EtOH:HMDS (1:1) 15 min at RT, in 100% HMDS for 15 min at RT, and lastly in 100% HMDS for 30 min at RT. HMDS was used to replace critical point drying. Samples were not metalized. SEM pictures were obtained using a *MiniMEB Hitachi TM3030* (University of French Polynesia). Given that lamellae are well formed and have a macroscopic size in *A. triostegus* at recruitment, lamellae counting was performed on tissues samples identically prepared, but observed under a simple binocular stereomicroscope. Lamellae counting was performed on both left and right olfactory organs to then have a measure of the average number of lamellae per olfactory organ. In the case where one of the left or right olfactory organ was damaged and thus accurate lamellae counts were not possible, we used the count from the other olfactory organ as the average number, as no significant difference was observed between the number of lamellae in the left and right olfactory organs in our preliminary observations.

**Lateral line trunk canal pores**. The lateral line system enables fish to detect water motions and pressure gradients, such as those caused by other fish (e.g., movement from other fish in the shoal or predator strikes). It is composed of superficial and canal neuromast receptor organs, which are the functional units of the lateral line system and are ciliary sensory organs, composed of hair cells, like those in the inner ear, located either on the skin or embedded in lateral line canals[43]. Canal neuromasts are found in the epithelium lining the bottom of the lateral line canals, and one canal neuromast is usually found within the short canal segments between two adjacent canal pores[43]. The development of neuromasts and the morphogenesis of lateral line canals and their pores initiate in late-stage larvae and continue through metamorphosis[43]. Counting the number of pores on the trunk canal is thus an appropriate way to rapidly characterize the development of the lateral line system when one cannot perform more advanced histological analyzes of the neuromasts. In d0 to d8 *A. triostegus* at recruitment, a fully formed trunk canal corresponding to a complete arched canal can be observed on each of the fish body flanks (Supplementary Fig. 4a). At this stage, *A. triostegus* does not exhibit scales[69], and the trunk canal is therefore only composed of soft tissue (Supplementary Fig. 4a).

Following the same preparation protocol as used for olfactory organs, we investigated the development of the lateral line system of *A. triostegus* by counting the number of pores on the trunk canal (Supplementary Fig. 4a-b). Counting was performed either on the left or right side of the body, depending on tissue preservation. Indeed, as fish body frequently curved when dried to a critical point in HMDS, only one side of the body was generally accessible for counting trunk canal pores. We did not consider this an issue, as preliminary observations revealed no significant difference between the numbers of trunk canal pores present on the left or right flanks of the body. Variation in the number of pores cannot be attributed to variation in the length of the fully formed trunk canal (measured from the vicinity of the head to the base of the tail) as it does not change in *A. triostegus* during metamorphosis (Supplementary Fig. 9).

**Thyroid hormone quantification**. TH was extracted from frozen fish, following an extraction protocol adapted from previous studies[70–72], including on coral-reef fishes[11]. Fish were individually crushed in 500 μl of methanol using a *Minilys* and glass beads (3*30 s, 5.000 rpm), centrifuged at 4 °C (10 min, 12.000 rpm), and supernatant reserved. These operations were performed twice, then fish were crushed one last time with 400 μl of methanol, 100 μl of chloroform, and 100 μl of barbital buffer (3*30 s, 5.000 rpm, RT), centrifuged at 4 °C (10 min, 12.000 rpm), and supernatant reserved. Pooled supernatants were then dried at 70 °C for 2 h. Dried pellets were re-suspended in 400 μl of methanol, 100 μl of chloroform, and 100 μl of barbital buffer, centrifuged at 4 °C (20 min, 12.000 rpm), and supernatant reserved. The same operation was again performed on the pellets. Pooled supernatants were once again dried out at 70 °C for 2 h with the final extract reconstituted in 2 ml of PBS 1× for quantification (*Roche Elica* kit on a *Cobas analyzer*, following the manufacturer's standardized method). TH levels in $pg\,g^{-1}$ of fish were then transformed into relative levels by selecting the respective control fish as standards in the increased temperature and CPF exposure treatment experiments.

**Behavioral tests**. A two-channel chemical choice flume[73–75] was used to assess the responses of *A. triostegus* towards chemical predator cues (Supplementary Fig. 10). Each trial presented an individual fish with two water sources: control seawater (Ø = UV-sterilized and 1-μm filtered seawater from the collection site) vs "seawater containing chemical cues from predator" (P = UV-sterilized and 1-μm filtered seawater from the collection site in which five *Lutjanus fulvus* predators were soaked, into a 125-L tank, for 2 h prior to the experiment). Water was fed into the flume through two water inlets, with equivalent flow rates of $100\,ml\,min^{-1}$ maintained using flow meters (MM Minimaster, *Admi-France*). This rate ensured laminar flow of each water source was maintained in parallel while allowing fish to swim naturally between the two water sources. Fine mesh and collimators also helped to ensure laminar flow of each water source was maintained[75]. Preliminary experiments were conducted to ensure the absence of unanticipated biased behavior within the two-channel flume (e.g., preference for one side of the flume over the other, irrespectively of the water sources, or preference for the drain area over the choice area). Prior to each trial, dye tests were conducted to confirm laminar flow, without eddies or areas of water mixing, within the choice area. After releasing the fish in the middle of the choice area, a 2-min acclimation period was observed, then fish position (left or right part of the choice area, or drain area) was recorded every 2 s for 5 min (Supplementary Fig. 10), using a camera (GoPro Hero 2) located above the edge of the flume tank. Water inlets were then switched (to account for any side preference due to fish's immobility) followed by another 2-min acclimation period. A second 5-min test period followed, during which fish position was again assessed every 2 s. Preference or avoidance of water sources, and the absence of a side preference, were confirmed by comparing responses during the first and second 5-min test periods. These experiments were performed in the dark with red light, to limit potential visual perturbations such as the presence of an observer and to allow comparison between d2 and d0 fish, as d0 fish were tested immediately after collection (i.e., at night) as this is when they are actively moving from the ocean to the reef, and is thus the most biologically relevant time to do so. Fish that did not swim actively during the first acclimation period were removed from the analysis ($n = 3$) to prevent side preference bias. However, all remaining fish swam actively between the two water sources after water inlets were switched (either during the second acclimation period or during the second test period), ensuring no continuous side bias due to immobility. Fish that spent more than 50% of the test time in the drain area (i.e., where the two water sources mix) were also removed from the analysis ($n = 4$) as we considered that they did not show any particular preference or avoidance for any of the two water sources. Fish that did not make a clear choice between the two water sources but spent more than 50% of the time in the choice area were included in the analysis. This was done as we wanted to assess fish preference as well as the absence of preference.

A double-choice tank was used to assess the responses of *A. triostegus* in the presence of a visual predator cues. Each trial presented an individual fish with a choice of two visual stimuli, each contained in a separate aquarium placed at the end of the central rectangular choice tank. These were an aquarium, containing a 10-cm (standard length) *L. fulvus* (P condition) vs an empty aquarium, equipped with an air stone (Ø condition) (Supplementary Fig. 11). Fish were first placed into the central "no choice" area of the choice tank for a 2-min acclimation period. During this time, fish were not able to see the contents of either adjacent aquaria or access the "choice" areas of the choice tank as opaque panels were

positioned at the edges of the "no choice" area (see the dotted lines in Supplementary Fig. 11). Following the acclimation period, the opaque panels were removed, allowing fish to observe the visual stimuli in the adjacent aquaria through the transparent walls of the choice tank and to access the choice areas. Fish position (i.e., choice area 1, no choice area, choice area 2; Supplementary Fig. 11) was then assessed every 2 s over a 10 min, using a camera (GoPro Hero 2) to limit any external visual disturbances such as an observer's presence. The camera was located above the choice tank. Location of visual stimulus (left or right side of the double-choice tank) was switched between each fish to ensure the absence of a side preference. Fish that remained immobile and spent >50% of the test time in the "no choice" area were removed from the analysis ($n = 11$), as we considered that they did not show any particular preference or avoidance for any of the two visual stimuli.

**Survival arena**. The survival tests presented in Figs. 2c, 3f were conducted in an arena set up in situ in a nursery lagoon area of the north coast (S17°29′7.0272″, W149°49′51.1166″). The arena consisted of a 1-m³ cage with a hard bottom covered with sand and coral rubble and four lateral walls made from 5-mm fine mesh (Supplementary Fig. 12). Each trial consisted of 45 fishes, with 15 from each treatment group. Fish from each treatment group were tagged with a specific color using visible implant fluorescent filament (Northwest Marine Technology) 2 h prior being released into the arena. For each trial, all fishes were released simultaneously and allowed to acclimate for 30 min before the introduction of six *L. fulvus* (15–20 cm SL). After 2 h, predators were removed, and surviving *A. triostegus* was identified to treatment using their color tag. Color tags attributed to each fish group were randomly switched between each replicate to ensure no predation bias based on tag color. Differences in survival during this predation test cannot be attributed to variation in fish size or condition between treatments, as these did not differ significantly between groups (Supplementary Fig. 13).

**Statistical analyses**. All statistical analyses were conducted using R version 3.5.3[76]. Conway–Maxwell–Poisson (COM-Poisson) generalized linear models (GLM) were used to assess if pharmacological and anthropogenic stressor treatments influenced the number of lamellae, the number of trunk canal pores, and the number of survivors to the predation experiment[77]. COM-Poisson GLM were also used to assess if pharmacological treatments influenced the number of photo-receptor external segments. Linear models (LM) were used to assess if pharmacological treatments influenced the bipolar cell, ganglion cell, and photoreceptor nuclei densities, and if pharmacological and anthropogenic stressor treatments influenced fish Fulton's K condition factor. LM were also used to assess if the trunk canal length varied with age. Gamma generalized linear mixed-effect models (GLMEM) were used to assess if anthropogenic stressor exposures influenced TH levels and $T_3/T_4$ ratios[78]. TH level or $T_3/T_4$ ratios were used as the dependent variable, and replicate was included as a random factor to account for differences in TH levels only due to the two different *Cobas* analyzers that were used in the two different years. As preliminary experiments provided no evidence that season and lunar phase affected $T_3$ levels in metamorphosing *A. triostegus*, we did not include them in our analyses (Gamma GLMEM, Supplementary Fig. 14). For each model, diagnostic plots were examined and outputs compared with raw data to confirm goodness-of-fit and residual homoscedasticity, and, when applicable, residual normality was assessed using Shapiro–Wilk normality test. Paired $t$ tests or Wilcoxon signed-rank tests were used to assess whether fish spent more time in the no cue choice area vs predator-cue choice area, depending on residual normality (Shapiro–Wilk normality test).

**Reporting summary**. Further information on research design is available in the Nature Research Reporting Summary linked to this article.

## Data availability
The source data underlying Figs. 1e–g, 2a–c, 3a–f, 4a, b, and Supplementary Figs. 1, 2, 6, 7, 8, 9, 13, and 14 are provided as a Source Data file. The unprocessed data are also available at: https://doi.org/10.5281/zenodo.3742258. Source data are provided with this paper.

## Code availability
R-code for statistical analyses and figure construction is available at: https://doi.org/10.5281/zenodo.3742258. Source data are provided with this paper.

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

## Acknowledgements

We thank Te-mana-o-te-moana, G. Iwankow, A. Morgan, C. Gache, L. Latry, J. Webb, and all CRIOBE and OOB people for their help in sampling, processing, or interpreting the data presented in this paper. The IAEA is grateful for the support provided to its Environment Laboratories by the Government of the Principality of Monaco. This research was supported by Agence Nationale de la Recherche (MANINI ANR19-CE34-0006-02) to D.L. and V.L., Contrat de Projet Etat-Polynésie française 2015-2020, LabEx Corail (Etape), PSL Environment (Pesticor), and Agence de l'eau (Rhone-Méditérranée-Corse n°2018-1765) to D.L., Fondation Bettencourt-Schueller to M.B., ANSES(2015/1/076) to V.L and by Griffith University and the Alexander von Humboldt Foundation to W.E.F.

## Author contributions

M.B., G.H., V.L., and D.L. designed the study; M.B., W.E.F., I.M., and L.F. collected and analyzed the data under the guidance of V.L. and D.L.; M.B. and W.E.F. wrote the paper with significant contributions from R.M.B., M.M., N.R., V.L., and D.L.

## Competing interests

The authors declare no competing interests
