## [Peer Review File · Nature Communications]

Reviewers' Comments:

Reviewer #1:

Remarks to the Author:

This paper, which follows on prior work by several of the authors, looks at the effect of temperature and a common pesticide on thyroid hormone levels in settlement stage larvae of a coral reef surgeonfish. Treatment with two stressors revealed reduction in thyroid hormone levels. Then effects of high and low thyroid hormone levels (treatments for increased and decreased levels via injection) were shown to have an effect on the "maturation" of three sensory systems – olfactory, visual and lateral line and, in turn, effects on predation vulnerability.

The study is well-constructed with appropriate controls (but no shams) and the statistical analysis appears to be appropriate. Use of a field site with the opportunity for in situ grow out of small fishes after treatments is unique.

Issues that need to be improved or reconciled:

1. The results on manipulation of thyroid hormone levels on sensory organ maturation are presented first, followed by results on the effect of introduced stressors on thyroid levels. To tell the story in a logical manner, perhaps this sequence can be reversed.

2. In presenting the results and their interpretation, the origin of supporting data from prior studies (e.g., two stressors together caused death, line 187, etc.) need to be more obvious to make the discussion of the data from this study more clear (check throughout).

3. Metamorphosis (morphological transformation), settlement (behavioral transformation) and recruitment (ecological process) are distinct from one another in fishes, but are often synchronized, or sequential (in rapid succession). However, it appears that these terms are used interchangeably in the manuscript – this needs to be fixed.

4. The morphology of the sensory organs (eye, nose) and system (lateral line; the organs are the neuromast receptor organs, which are not discussed at all) are described inaccurately or are not sufficiently described. Please also check all figure captions including captions in the supplementary materials for accuracy of terminology (see below).

The olfactory organ is a fluid filled blind sac that contains the ciliated sensory epithelium that forms a rosette comprise of a number of lamellae (feel free to use this text). In the MS – the terms nostril, rosetta are not used correctly. Line 100 – "nostril lamellae" – use just "lamellae". Line 375 – "each nostril is covered by an olfactory epithelium" should be "each olfactory organ contains an olfactory epithelium. Line 379 – "olfactory organ cavity" should be "olfactory organ".

The retina is composed of rods and cones, two types of retinal photoreceptors (not photcones, check throughout).

The portion of the lateral line system examined in the study is limited to the fully-formed trunk canal (found in most post-metamorphic fishes = juveniles). It would have been helpful to know the morphological condition of the lateral line canals on the head as well - were they informative?.

Counting the pores associated with the trunk canal (not a lateral line) is appropriate, however I suggest using a graphic icon that is more stylized. The icon used is not recognizable as a component of the lateral line system and was confusing. On line 101, it is stated that there is a rapid surge in lateral line canal pore density, but it is not clear what a rapid surge is, and how this compares to other species for instance. Finally, lateral line length, a parameter analyzed in this study, must be defined.

5. No mention is made about the mechanism underlying the effect of thyroid hormones on the morphology of the particular sensory systems examined (neurogenesis vs. epithelial proliferation).

This is important since the results for the olfactory system in response to the pesticide are different from that for the eye and the lateral line system. It should be noted that the development of the nose, eye and the lateral line system are relatively gradual through the larval stage of fishes with some morphological transformations that occur at metamorphosis (enclosure of nose, formation of lateral line canals, both of which are complete in the fishes used in this study, which is interesting).

Comparisons with other well studied fish species (ideally coral reef fishes, or other marine fishes with a pelagic larva) would have been helpful in this context.

Olfactory – the number of lamellae (please check method used in the literature for counting bilaterally symmetrical lamellae in the rosette; however, this should not affect the trends in the data as presented) is roughly correlated with the number of olfactory neurons that are the receptor cells. This reflects a process of neurogenesis. Histology (tissue sectioning) would have more directly revealed numbers of receptor cells, which would have been more indicative of a process of functional relevance that is being altered by thyroid and stressor treatments. In larval fishes, the olfactory epithelium is exposed on the surface of the “snout” and (typically at metamorphosis), the epithelium sinks into a blind fluid filled sac and nostrils (nares) form in that process. The fact that so many lamellae are present at the time of collection (“settlement stage” larvae, d0) suggests that olfactory development is advanced at this stage compared to other coral reef fishes (see Hu et al. 2019, in J. Fish Biol; Lara – work on nose of wrasses – cited in Hu et al.)

Eye/retina – maturation of the retina (Number and relative thickness of layers) should be compared to that any well-studied fish, for instance, zebrafish. Comparison to a goatfish (line 351) which is a benthic feeding the fish may not be appropriate when comparing the ventral versus the dorsal sides of the retina. Further, measuring densities at in the peripheral area of the retina may not be accurate for the reason that is stated - cell densities are higher in the central area of the retina - but densities in this area maybe more indicative of visual abilities. The literature should be checked on this point.

Lateral line – unlike the measurements taken for the nose and the retina the morphological parameter used to evaluate lateral line maturation (the density of canal pores) is not a function of the process of neurogenesis (see line 81), rather it is a process of enclosure any elaboration of the canal itself. It would have been helpful to know if the canal is found with in the lateral line scales or if the canal is composed only of soft tissue. Thus, maturation in the three sensory organs as defined in the study are not equivalent. This needs to be acknowledged. That being said, the addition of pores is a legitimate measure of the “maturation” of lateral line canal on the trunk, as it would be for canals on the head.

6. More specific issues:

Line 108 – change nostril to olfactory organ, and change lateral line to trunk canal and correct this throughout.

Line 114 – what study looked at olfactory and visual cues, and is it known that those cues were olfactory and not gustatory. If this is not known please use “chemical cue” or “chemical choice experiment” and not “olfactory cue” or “olfactory choice experiment.” The same arguments can be used for the term “olfactory” on line 432 unless the cited papers have clearly demonstrated that the behavior is guided by olfaction and not gustation.

Line 205 & 210 – “anthropogenised” is not a word. Please change this.

Line 214 – the argument for the fitness consequences needs better support with citations.

Line 218 – please define “replenishment potential”.

Line 247 – please more precisely define “settlement stage”. What was the condition of the other markers of metamorphosis such as meristic counts, presence of scales, and transitioning or juvenile pigmentation.

Line 335 – the use of the term histology is correct here, but because tissue sectioning (the more common definition of histology) is not involved please remove the term histology from lines 371 and 398.

Line 401 – There are newer reviews of lateral line structure and function that should be cited here. For instance the 2014 volume called “the lateral line system” (Springer).

Reviewer #2:

Remarks to the Author:

In this manuscript, Besson and colleagues demonstrate that disrupted TH signaling leads to aberrant development of sensory structures in *A. triostegus*, and that blocking TH signaling impairs the ability of fish to avoid predators. The authors then show that increased temperatures or CPF exposure during metamorphosis lower T3 and T4 levels, impair development of sensory structures and decrease survival in a predation assay. Survival in the predation assay is rescued by treatment with T3, and the authors conclude that TH disruption is the causal mechanism underlying decreased survival in fish exposed to higher temperatures or CPF. Additional data would strengthen this causal link, and the experiment needs an additional control (CT injected with T3). Especially without data showing that the T3 treatment rescues sensory structure development or avoidance behavior, the authors should explore alternative explanations for their increase in survival in T3-rescued fish. However, overall, these findings are novel, timely and important, the manuscript is well written, and the data are presented beautifully.

My specific critiques, in no particular order:

Some citations are repeated (9 and 53)

Line 81 should include the citation Hu et al 2019 Dev Dyn, which shows that TH is essential for development of the lateral line system in zebrafish. The lateral line results should be compared to the results in Hu et al.

The introduction and discussion would be strengthened by mentioning that aquatic organisms are even less buffered against climate change than terrestrial organisms, citing work including Pinsky et al 2019.

In Fig 2, it is unclear to me why there seems to be a difference in the time spent at stimulus area with no cue between the control and T3 treated fish? Is this apparent difference significant and can it be addressed?

In lines 183-18, the authors say that temperature has a higher impact on T4 levels than CPF, and this is a bit unclear to me. It looks in the figure like the +3°C treatment results in a similar T4 level as CPF30. Do the authors mean that the ratio of T4/T3 is different between the different treatments? My eye is not detecting ratio differences either. Can this be clarified? If the goal is to highlight differences in T3/T4 ratios, it would be very helpful to show a graph with T3/T4 ratios for each individual in each treatment.

Lines 190-192, the authors say that exposure to both +3°C and CPF5 or CPF30 causes death. I do not see these data presented in the manuscript, and the citation listed (Wu et al 2017) does not contain that experiment. This needs to be clarified.

The authors show in Fig 3F that the survival of fish exposed to high temperatures or to CPF can be rescued by T3 injections. This is a very exciting finding, but it left me wondering about the actual behavior of these rescued fish and the development of their sensory organs. Is it possible to show data for the rescued fish in some or all graphs in Fig 1, 2 and Fig 3 C-D? Without showing these data, it is difficult to establish a mechanistic link between the T3 rescue and the increase in survival. Further, are there data for a T3 treatment on control fish? Do these show any difference in survivorship from euthyroid controls? (Also, do they show differences in development of sensory

organs or predator avoidance?) This control is essential in interpreting the findings. My lab has found that hyperthyroid fish are generally hyperactive and tend to swim very fast and erratically (this is an unpublished observation). I wonder if the rescue you see in survival rates is due to the hyperthyroid fish swimming faster and more erratically, rather than actually having properly developed sensory structures and rescued appropriate predator avoidance. Alternatively, is it possible that the T3 treated fish smell or taste different to the predators, and could that explain the decrease in predation? These possibilities should be explicitly tested or at least addressed as possibilities.

-Sarah McMenamin

Reviewer #3:

Remarks to the Author:

Review of manuscript: Anthropogenic stressors undermine fish sensory development and survival via thyroid disruption. In this manuscript, the authors aim to investigate the physiological mechanism underlying observed effects of anthropogenic stressors on fish behaviour and survival. They tested two stressors: increased temperature and a chemical stressor (chlorpyrifos pesticide), and investigated the effect of these on the metamorphosis stage using the convict surgeonfish (*Acanthurus triostegus*). Metamorphosis is controlled by thyroid-hormones (TH), wherefore the effects of these stressors on TH levels was investigated. They also investigated the effect of TH levels and the two stressors on predator avoidance behaviour and survival. The authors report that both stressors decreased TH levels and increased predation vulnerability.

The manuscript is impressive in terms of containing a lot of data. The experimental design is thorough, with the authors first investigating sensory organ maturation and the role of TH during metamorphosis (with the TH signal disruption treatment, N3, showing repressed sensory organ maturation), then the effect of TH on behaviour (with N3 treated fish showing no response to chemical or visual cues from a predator, which led to lower survival). These experiments were followed by investigations of the effects of the stressors (separately and combined) on TH signalling and sensory development, and whether TH injections can restore survival rates. Several treatment levels were used, in combination with controls and solvent controls, as well as some experimental sampling during different days of the metamorphosis. This also means that there are many different treatments to keep track of for the reader. The authors did a pretty good job at presenting the data in a clear and concise manner, but due to the high number of treatments it is still sometimes difficult to follow all results.

The experimental design is quite well described, and I commend the authors for being clear, for example on number of replicates removed and which criteria that were used for removing those (e.g. page 18-19, line 544-464, line 480-482)). However, several important method aspects are missing, such as which year each experiment was performed (coupled with a lack of testing for year effect in the statistical analyses), how many in situ cages and aquaria that were used for housing and exposing the fish (coupled with a lack of tank-effect analysis), and in particular method description and data for how the warming and chemical treatments were maintained and measured. I did not find any description or data of analysis of water and fish samples from the chemical treatment to verify that the targeted levels were reached? See specific comments below. Also, the statistical methods used are rather simple, and while complicated statistical tools should not be a target in itself, more modern tools could be more appropriate, for example rather than using non-parametric tests, which was used in the manuscript for certain analyses.

Another general comment is the overall emphasis on the negative effects found. In particular in the title and abstract, but in general the paper gives the impression that these stressors had a negative impact on sensory development and behaviour, period. However most of the effects were seen in the most extreme treatments, i.e. +3°C and the CPF 30 µg/L treatment (e.g. as shown in Fig. 3), while the lower temperature (+1.5°C) and the lower levels of CPF (1 and 5 µg/L) only had effects on some of the measurements, effects that were not consistent. The fact that these treatments had no effect is mentioned in the result section, but largely ignored in the rest of the manuscript. The authors need to make it clear that the lower treatments had none, or a smaller and non-consistent effects.

I also have some concerns regarding the choice of the targeted chemical levels. The chlorpyrifos (CPF) levels used were 1, 5, and 30 µg/L. These levels are argued to be biologically relevant in the discussion (page 8, line 197-199). However, the cited report (reference 45, NRA Review of Chlorpyrifos) describes that chlorpyrifos is an occasional contaminant of surface waters, and that levels are usually below 1 µg/L. These values are from rivers, which would be relevant if tested on a freshwater organism inhabiting such areas. Since this manuscript use a marine species, levels measured in the sea where they occur are needed in order to say what is biologically relevant. If levels are usually below 1 µg/L in rivers, then those values would be much lower in the oceans due to the dilution effect. The report state that only a few high outliers have been measured, where levels reached 25-26 µg/L, this was again in rivers and irrigation drainage. Based on this, I would say that even the lowest level used here is not biologically relevant, and the highest level used is even higher than extreme outliers. The results are still interesting, knowing the effects of high levels of contaminants can be important, for example it is highly likely that levels will be higher in the future. However it should not be called biologically and/or environmentally relevant. Regardless the levels used here, any values of this contaminant measured closer to where the fish were collected, or at least in a coral reef environment, would be much more relevant to cite. The other references used by the authors to justify the levels used are for example reference 46 (Bigot et al. 2016), however they report levels of 0.18-0.54 pg/L, i.e. magnitudes lower than what was used here. Reference 57 (John & Shaike 2015) does not seem to include much data on environmentally relevant levels, but for example discuss toxicity and LC50 values. Reference 13 (Besson et al. 2017) do state that Australian reef surface waters can reach up to 1 µg/L, but I was unable to find the paper that is cited for this value (NRA. in National Registration Authority for Agricultural and veterinary Chemicals 1, 17 2000), meaning that the validity of this data point cannot be checked or confirmed. In relation to this, it could also be discussed how biologically relevant a 36h (page 13, line 308-309) chemical exposure period is?

Overall I think the manuscript has good potential to be of high value to many fields, but these concerns must be clarified first.

Specific comments:

Was any data collected blind regarding treatment? I could not find such a statement so I assume it was not. This is fine, sometimes blind data collection is impossible, however this should be stated in the paper. In particular given the choice flume experiment was not video recorded despite only lasting rather short period of time (14 min for the choice flume experiment, out of which 10 min were used for data collection), and the apparent availability of a camera, which was used in the second behavioural experiment testing visual cues (page 19, line 477-478).

Throughout the manuscript, please change the word "olfaction", unless you are certain that this is the sensory system under investigation (which most of the time you are not, since the other sensory systems (chemoreception and taste) were not blocked). Use the word "chemosensory cues", or similar.

Page 10, line 246: The study was conducted over several years (2015-2018). Which experiment was

conducted when? Could there be differences between years? This was not statistically tested.

Page 11, line 249-252: How many cages were used for the thyroid hormone signalling experiment? On line 277 it says 15 fish were kept in each in situ cage, but not how many cages that were used. Similarly, for the temperature and chlorpyrifos exposure experiments, how many aquaria were used? This also needs to be incorporated in the statistical analyses.

Page 12, line 290-296: Please describe the method used to increase temperature, how it was maintained and controlled. Please include data on actual measured temperature per tank and treatment. How was the increase applied? I.e., using some ramping protocol? Or were the fish put straight from 28.5°C into the warm treatments?

Page 13, line 300-303: Please add methods describing how the chemical treatments were obtained (e.g., was the chemical simply mixed into each tank together with solvent?). In particular, give methods for how the target concentrations were measured and controlled. Please include data on actual measured concentrations per tank and treatment. In ecotoxicology it is standard that not only the treatment water concentrations are analysed, but also fish tissue. This was not done here (and the way the methods are written, it is not even given that water concentrations were measured). This means that the actual concentrations in the fish are unknown, and also so in the water?

Page 17, line 434-437: Please add the water volume in which the five predators were soaked for 2 h to create the predator odour. The concentration of predator cue cannot be estimated based on the given information (meaning the experiment could not be repeated for this particular point).

Page 18, line 454-455: Are there any implications of testing a diurnal species at night? Red light was used, which is good, but the fish would not be expected to be active during night? In particular given they are sensitive to light pollution (artificial light at night, as stated on page 10, line 236-238). This should at least be discussed.

Page 19, line 477-478: For the second behavioural experiment, testing visual cues, the experiment was video recorded (please add details on camera used, placement of camera etc., and how the videos were analysed, was some software used?) in order to limit disturbance by an observer. However, in the choice flume experiment, an observer was present (page 18, line 453). How did you ensure that the observer did not cause any disturbance in the first experiment, when this was a concern in the second behavioural experiment?

Page 20, line 492-493: In my experience, obtaining an ethical permit to perform experiment where a prey fish is to be consumed by a predator is very difficult. In particular if the experimental design means putting prey and predators together and counting the number of surviving prey after a certain amount of time, as done here. Usually such experiments, if allowed at all, require constant monitoring, so that harmed fish can be removed and euthanized to prevent hours of suffering. Please indicate clearly that the experimental design used here was specifically included in the ethical permits and approved, for example by enclosing the ethical permit where this design is mentioned (the ethical permits given on page 10, line 223-225 are not public documents and hence there is no possibility to review this).

Page 20, statistical analysis: The fish were housed in groups in aquaria or in situ tanks, but there is no mention of checking for tank effects? There's also no analysis of differences between years, and information on which experiment that was conducted what year is missing.

Reviewer #1

General comment:

This paper, which follows on prior work by several of the authors, looks at the effect of temperature and a common pesticide on thyroid hormone levels in settlement stage larvae of a coral reef surgeonfish. Treatment with two stressors revealed reduction in thyroid hormone levels. Then effects of high and low thyroid hormone levels (treatments for increased and decreased levels via injection) were shown to have an effect on the “maturation” of three sensory systems – olfactory, visual and lateral line and, in turn, effects on predation vulnerability. The study is well-constructed with appropriate controls (but no shams) and the statistical analysis appears to be appropriate. Use of a field site with the opportunity for in situ grow out of small fishes after treatments is unique.

R: We thank the reviewer for their considered and thorough comments, and are confident that they have greatly helped us to improve the manuscript.

Specific comments:

1. The results on manipulation of thyroid hormone levels on sensory organ maturation are presented first, followed by results on the effect of introduced stressors on thyroid levels. To tell the story in a logical manner, perhaps this sequence can be reversed.

R: We appreciate the logic of this suggestion. However, as the discovery that thyroid hormones (TH) play a role in regulating recruitment processes in coral reef fishes is relatively new discovery (Holzer et al. 2017 – eLife 6:e27595), we believe that our message is best conveyed by first examining the role of TH signaling in sensory maturation and survival. We then extend on this to explore whether anthropogenic stressors can disrupt TH signaling with consequences for sensory maturation and survival.

2. In presenting the results and their interpretation, the origin of supporting data from prior studies (e.g., two stressors together caused death, line 187, etc.) need to be more obvious to make the discussion of the data from this study more clear (check throughout).

R: We agree and have edited the text accordingly. For example, line 187 (now 231-233) now reads:

“Given that stressors are rarely experienced individually, these results highlight the vulnerability of aquatic organism endocrine functions and the risk posed by anthropogenic stressors^{7,9}”

Concerning the presentation and interpretation of our results, we initially thought that referring to our figures in the Discussion was not necessary. However, we think this is a valuable point, and have followed the reviewer’s remark and we have edited the Discussion section so that the origin of supporting data is now clearly referenced (either with a Figure call or a literature reference).

3. Metamorphosis (morphological transformation), settlement (behavioral transformation) and recruitment (ecological process) are distinct from one another in fishes, but are often synchronized, or sequential (in rapid succession). However, it appears that these terms are used interchangeably in the manuscript – this needs to be fixed.

R: We thank the reviewer for pointing out that we mistakenly used the terms metamorphosis and recruitment interchangeably in the previous version of the manuscript. We have modified the text so that we now use the term “metamorphosis” to refer exclusively to the developmental process and we use the term “recruitment” to refer exclusively to the ecological process. Regarding “settlement”, this term is widely used to refer to the precise period of time when larval fish move from the ocean to the reef (i.e. reef entry or reef colonization) rather than referring to the behavioral transformations that occur at this stage (see Atema et al. 2002 MEPS 241: 151-160; Leis et al. 2002 MEPS 232: 259-268; Wright et al. 2010 Coral Reefs 29: 235-243; Sponaugle et al. 2012 MEPS 453: 201-212). We have made the appropriate changes throughout the manuscript to make sure that the use of the term settlement is consistent with the above-mentioned meaning.

4. The morphology of the sensory organs (eye, nose) and system (lateral line; the organs are the neuromast receptor organs, which are not discussed at all) are described inaccurately or are not sufficiently described. Please also check all figure captions including captions in the supplementary materials for accuracy of terminology (see below). The olfactory organ is a fluid filled blind sac that contains the ciliated sensory epithelium that forms a rosette comprise of a number of lamellae (feel free to use this text). In the MS – the terms nostril, rosetta are not used correctly.

R: We are very grateful for the detailed comments provided by this reviewer, and we have modified the manuscript (including figures and captions) and supplementary material accordingly. For example, the text in our Methods section that describes the olfactory organ now reads (lines 442-446):

“In fish, the olfactory organ is a fluid filled blind sac that contains the ciliated sensory epithelium that forms a rosette comprise of a number of lamellae⁴⁴. In *A. triostegus*, the left and right olfactory organs can be found in two cavities on the dorsal surface, between the eye and the snout edge, with water circulating in each cavity from the anterior nostril to the posterior nostril (Supplementary Fig. 3).”

Concerning the lateral line and trunk canal, we have added the following information to the Methods regarding the neuromast receptor organs, the trunk canal and its pores (lines 468-482):

“The lateral line system enables fish to detect water motions and pressure gradients, such as those caused by other fish (e.g. movement from other fish in the shoal or predator strikes). It is composed of superficial and canal neuromast receptor organs, which are the functional units of the lateral line system and are ciliary sensory structures located either on the skin or embedded in lateral line canals⁴³. Canal neuromasts are found in the epithelium lining the bottom of the lateral line canals, and one canal neuromast is usually found between two adjacent canal pores (e.g. on the cranial lateral lines) or at the level of the canal pore (e.g. in trunk canals)⁴³. Neuromast maturation and morphogenesis of lateral line canals and their pores initiate in late-stage larvae and continue through metamorphosis⁴³. Counting the number of pores on the trunk canal is thus an appropriate way to rapidly characterize the maturation of the lateral line system when one cannot perform more advanced histological analyzes of the neuromasts. In d0 to d8 *A. triostegus* at recruitment, a fully formed trunk canal corresponding to a complete arched canal can be observed on each of the fish body flanks (Supplementary Fig. 4a). At this stage, *A. triostegus* also only exhibits very thin calcified vertical plates but no scales⁶⁹, and the trunk canal is therefore only composed of soft tissue (Supplementary Fig. 4a). Following the same preparation protocol as used for olfactory organs, we investigated the maturation of the lateral line system of *A. triostegus* by counting the number of pores on the trunk canal (Supplementary Fig. 4a-b).”

5. Line 100 – “nostril lamellae” – use just “lamellae”.

R: Agreed. We have modified the text accordingly.

6. Line 375 – “each nostril is covered by an olfactory epithelium” should be “each olfactory organ contains an olfactory epithelium.”

R: Agreed. This was addressed in response to comment #4 and changed accordingly.

7. Line 379 – “olfactory organ cavity” should be “olfactory organ”.

R: Agreed. We have modified the text accordingly.

8. The retina is composed of rods and cones, two types of retinal photoreceptors (not photocones, check throughout).

R: Thank you for this comment. We have replaced “photocones” with “photoreceptors” throughout the text.

9. The portion of the lateral line system examined in the study is limited to the fully-formed trunk canal (found in most post-metamorphic fishes = juveniles). It would have been helpful to know the morphological condition of the lateral line canals on the head as well - were they informative?

R: Unfortunately, the lateral line canals on the head were not available for analysis as we used fish heads to study the retina and/or olfactory organ. We have modified our manuscript to fully mention that only trunk canal pores were observed. For example, we have replaced “lateral line pores” with “trunk canal pores” throughout the text. Please see also our response to comment #4 for further relevant text revision.

10. Counting the pores associated with the trunk canal (not a lateral line) is appropriate, however I suggest using a graphic icon that is more stylized. The icon used is not recognizable as a component of the lateral line system and was confusing.

R: The icon used is based directly on the SEM image provided in Fig. 1d. This is also the case for the graphic icon used for the bpc density (based on Fig. 1c) and the lamellae (based on Fig. 1b). Therefore, we cannot think of a better way to highlight this structure, and would prefer to keep the graphic as it is. That being said, we have modified our figure caption to more explicitly highlight the link between Fig. 1a, Fig. 1d and Supplementary Fig. 4 (lines 819-820):

“SEM of a trunk canal pore (demarcated by a blue-line). Several trunk canal pores are found in the blue dotted-area in the panel a (Supplementary Fig. 4).”

11. On line 101, it is stated that there is a rapid surge in lateral line canal pore density, but it is not clear what a rapid surge is, and how this compares to other species for instance.

R: We agree that we could have been clearer here, and have modified the text to be more precise. For example, lines 105-108 now read:

“The olfactory, visual, and mechanosensory organs (Fig. 1b-d) showed rapid maturation from d0 to d8 in control (CT) individuals, with the development of new lamellae, a 50% increase in bipolar cell (bpc) density in the retina, and a 240% surge in trunk canal pore density, respectively (Fig. 1e-g).”

Concerning the comparison with other species, lines 186-191 now read:

“*A. triostegus* exhibits an enclosed nose (Supplementary Fig. 3a), well-formed rosette lamellae (Supplementary Fig. 3b) and a fully formed trunk canal (Supplementary Fig. 4a) at the time of settlement, but relatively low cell densities in the retina (Fig. 1F and Supplementary Fig. 6-8)^{13,39}. This suggests that, with the possible exception of the retina, the sensory structures of *A. triostegus* are generally more developed than other reef⁴⁰⁻⁴² and non-reef^{36,43} fish species at a similar life-history stage.”

12. Finally, lateral line length, a parameter analyzed in this study, must be defined.

R: This parameter is now defined in the text on lines 489-491:

“Variation in the number of pores cannot be attributed to variation in the length of the fully formed trunk canal (measured from the vicinity of the head to the base of the tail) as it does not change in *A. triostegus* during metamorphosis (Supplementary Fig. 9).”

13. No mention is made about the mechanism underlying the effect of thyroid hormones on the morphology of the particular sensory systems examined (neurogenesis vs. epithelial proliferation). This is important since the results for the olfactory system in response to the pesticide are different from that for the eye and the lateral line system. It should be noted that the development of the nose, eye and the lateral line system are relatively gradual through the larval stage of fishes with some morphological transformations that occur at metamorphosis (enclosure of nose, formation of lateral line canals, both of which are complete in the fishes used in this study, which is interesting). Comparisons with other well studied fish species (ideally coral reef fishes, or other marine fishes with a pelagic larva) would have been helpful in this context.

R: It would be very interesting to disentangle and decipher the mechanisms underlying the effects of thyroid hormones on different sensory systems. While this was not the focus of our study, we agree that this would make for a great (albeit challenging) follow-up study. We have highlighted this in the discussion, and have made comparisons with other coral reef fishes (lines 184-204):

“Thyroid hormones are important for regulating sensory development in teleost fishes^{11,14,36,38}, and our results provide insights into the role they play in sensory system maturation during recruitment. *A.*

triestegus exhibits an enclosed nose (Supplementary Fig. 3a), well-formed rosette lamellae (Supplementary Fig. 3b) and a fully formed trunk canal (Supplementary Fig. 4a) at the time of settlement, but relatively low cell densities in the retina (Fig. 1F and Supplementary Fig. 6-8)^{13,39}. This suggests that, with the possible exception of the retina, the sensory structures of *A. triestegus* are generally more developed than other reef⁴⁰⁻⁴² and non-reef^{36,43} fish species at a similar life-history stage. Despite this advanced stage of development, we found that fish with pharmacologically promoted TH signaling experienced faster sensory organ maturation, those with disrupted TH signaling (either pharmacologically (Fig. 1e-g) or environmentally (Fig. 3c,e)) experienced impaired sensory development, and those with pharmacologically disrupted TH that received supplemental T₃ experienced rescued maturation of their olfactory organ (Supplementary Fig. 1). This development of new lamellae promoted by TH is consistent with neurogenesis³⁵, as the number of lamellae roughly correlates with the number of olfactory neurons⁴⁴. In contrast, the maturation of the trunk canal involves both the development of new canal neuromasts (i.e. neurogenesis) and the enclosure and elaboration of the canal itself⁴³, which is more consistent with epithelial proliferation. The fact that lamellae development was not affected by increased temperature or CPF (Fig. 3d) suggests that the endocrine disruption caused by these stressors may not be severe enough (e.g. in comparison with the N3 treatment) to affect this maturation process. These results highlight the sensitivity and complexity of the mechanisms underlying the actions of TH on sensory system maturation, offering an avenue for future research.”

14. Olfactory – the number of lamellae (please check method used in the literature for counting bilaterally symmetrical lamellae in the rosette; however, this should not affect the trends in the data as presented) is roughly correlated with the number of olfactory neurons that are the receptor cells. This reflects a process of neurogenesis.

R: We confirm that the methods that we used are identical to those used in previous landmark studies in this field (Kasumyan 2004 J Ichthol 44: 180–223; Pashchenko & Kasumyan 2015 J Ichthyol 55: 880–899; Ghosh & Chakrabarti 2016 Mesopotamian Journal of Marine Science 31(1) 15-28; Pashchenko & Kasumyan 2017 J Ichthyol 57: 136–151). Yes, the number of lamellae should roughly correlate with the number of olfactory neurons in a receptor cell. This suggests that our results are consistent with neurogenesis (see our reply to comment #13), and this is consistent with the well-known effect of thyroid hormones on neurogenesis in a number of organisms (Gothié et al., 2017 Mol Cel Endocrinol, 459, 104-115, which we cite in both the Introduction and Discussion).

15. *Histology (tissue sectioning) would have more directly revealed numbers of receptor cells, which would have been more indicative of a process of functional relevance that is being altered by thyroid and stressor treatments. In larval fishes, the olfactory epithelium is exposed on the surface of the “snout” and (typically at metamorphosis), the epithelium sinks into a blind fluid filled sac and nostrils (nares) form in that process.*

R: We agree that tissue sectioning could have been a great complementary approach to study the maturation of the olfactory organ. That being said, our results do show that TH signaling affects the development of new lamellae, which we believe is sufficient for this particular study.

16. *The fact that so many lamellae are present at the time of collection (“settlement stage” larvae, d0) suggests that olfactory development is advanced at this stage compared to other coral reef fishes (see Hu et al. 2019, in J. Fish Biol; Lara – work on nose of wrasses – cited in Hu et al.).*

R: We thank the reviewer for these references, and have added a paragraph discussing the sensory development of *A. triostegus* compared to other coral reef fishes (see our reply to comment #13), where we cite these references. In particular, lines 186-191 now read:

“*A. triostegus* exhibits an enclosed nose (Supplementary Fig. 3a), well-formed rosette lamellae (Supplementary Fig. 3b) and a fully formed trunk canal (Supplementary Fig. 4a) at the time of settlement, but relatively low cell densities in the retina (Fig. 1F and Supplementary Fig. 6-8)^{13,39}. This suggests that, with the possible exception of the retina, the sensory structures of *A. triostegus* are generally more developed than other reef⁴⁰⁻⁴² and non-reef^{36,43} fish species at a similar life-history stage.”

17. *Eye/retina – maturation of the retina (Number and relative thickness of layers) should be compared to that any well-studied fish, for instance, zebrafish. Comparison to a goatfish (line 351) which is a benthic feeding the fish may not be appropriate when comparing the ventral versus the dorsal sides of the retina. Further, measuring densities at in the peripheral area of the retina may not be accurate for the reason that is stated - cell densities are higher in the central area of the retina - but densities in this area maybe more indicative of visual abilities. The literature should be checked on this point.*

R: We agree that our results should be compared to ecologically similar species, and we would like to highlight that both *Upuneus tragula* (goatfish) and post-larval/juvenile stage *A. triostegus* are benthic feeders. While there are differences between the species (e.g. post-larval and juvenile *A. triostegus* live in rubble-dominated environments, while *U. tragula* prefer sandy areas), both species are marine, tropical, and reef-associated fishes

that present similar life cycles, undergo metamorphosis at recruitment, and spend the majority of their time at the bottom of the water column. We therefore believe that the comparison we make with *U. tragula* is more relevant than a comparison with *Danio rerio* (zebrafish) (freshwater species, occupying different habitat, not benthic, does not undergo important metamorphic changes as those undergone by reef fishes at the time of recruitment; typical laboratory model fish).

Regarding the second part of this comment #17, we are aware that cell densities could vary across the retina. However, our goal was to look at the role of thyroid hormones in the maturation of the retina, and we were only interested in the relative – not absolute – maturation of this organ. That being said, a recent study on *Naso brevirostris* (another Acanthuridae species) showed very weak, if any, spatial specialization in the retina of settlement-stage individuals (Tettamanti et al. 2019 J. Exp. Biol. 222, jeb209916). This weak pattern of spatial specialization was actually only observed on the temporal/nasal axis, not on the dorsal/ventral axis. Therefore, while we maintain that our methods are appropriate for our aim, we have modified our Methods text to ensure that another reader does not have the same question (lines 426-432):

“Also, we only examined the dorsal side (ds) of the retina, as the ventral side (vs) was shown to not undergo maturation at metamorphosis in another coral reef fish species, the goatfish *Upeneus tragula*¹³ (Supplementary Fig. 5a). To compare the maturation state of the retina between treatments, we looked at the peripheral area of the ds (see the dotted-square in Supplementary Fig. 5a, magnified in Supplementary Fig. 5b) as settlement-stage individuals in another Acanthuridae species (*Naso brevirostris*) showed weak if any spatial specialization of the retina, in particular on this axis⁶⁸.”

We now also discuss the maturation of the retina in *A. triostegus* compared to other coral reef fish species (lines 186-191):

“*A. triostegus* exhibits an enclosed nose (Supplementary Fig. 3a), well-formed rosette lamellae (Supplementary Fig. 3b) and a fully formed trunk canal (Supplementary Fig. 4a) at the time of settlement, but relatively low cell densities in the retina (Fig. 1F and Supplementary Fig. 6-8)^{13,39}. This suggests that, with the possible exception of the retina, the sensory structures of *A. triostegus* are generally more developed than other reef⁴⁰⁻⁴² and non-reef^{36,43} fish species at a similar life-history stage.”

18. Lateral line – unlike the measurements taken for the nose and the retina the morphological parameter used to evaluate lateral line maturation (the density of canal pores) is not a function of the process of neurogenesis (see line 81), rather it is a process of enclosure any elaboration of the canal itself. It would have been helpful to know if the canal is found within the lateral line scales or if the canal is composed only of soft tissue. Thus, maturation in the three sensory organs as defined in the study are not equivalent. This needs to be

acknowledged. That being said, the addition of pores is a legitimate measure of the “maturation” of lateral line canal on the trunk, as it would be for canals on the head.

R: Past work (e.g. Frédérich et al. 2010, J. Appl. Ichtyol. 26, 176-178) and our Supplementary Fig. 6A both show that the trunk canal is only composed of soft tissue at settlement. To make this clearer in the manuscript, we have modified the text (lines 478-482):

“In d0 to d8 *A. triostegus* at recruitment, a fully formed trunk canal corresponding to a complete arched canal can be observed on each of the fish body flanks (Supplementary Fig. 4a). At this stage, *A. triostegus* also only exhibits very thin calcified vertical plates but no scales⁶⁹, and the trunk canal is therefore only composed of soft tissue (Supplementary Fig. 4a).”

We also provide supplementary discussion concerning the maturation of sensory structures and acknowledge that their respective maturation is not equivalent (lines 184-204):

“Thyroid hormones are important for regulating sensory development in teleost fishes^{11,14,36,38}, and our results provide insights into the role they play in sensory system maturation during recruitment. *A. triostegus* exhibits an enclosed nose (Supplementary Fig. 3a), well-formed rosette lamellae (Supplementary Fig. 3b) and a fully formed trunk canal (Supplementary Fig. 4a) at the time of settlement, but relatively low cell densities in the retina (Fig. 1F and Supplementary Fig. 6-8)^{13,39}. This suggests that, with the possible exception of the retina, the sensory structures of *A. triostegus* are generally more developed than other reef⁴⁰⁻⁴² and non-reef^{36,43} fish species at a similar life-history stage. Despite this advanced stage of development, we found that fish with pharmacologically promoted TH signaling experienced faster sensory organ maturation, those with disrupted TH signaling (either pharmacologically (Fig. 1e-g) or environmentally (Fig. 3c,e)) experienced impaired sensory development, and those with pharmacologically disrupted TH that received supplemental T₃ experienced rescued maturation of their olfactory organ (Supplementary Fig. 1). This development of new lamellae promoted by TH is consistent with neurogenesis³⁵, as the number of lamellae roughly correlates with the number of olfactory neurons⁴⁴. In contrast, the maturation of the trunk canal involves both the development of new canal neuromasts (i.e. neurogenesis) and the enclosure and elaboration of the canal itself⁴³, which is more consistent with epithelial proliferation. The fact that lamellae development was not affected by increased temperature or CPF (Fig. 3d) suggests that the endocrine disruption caused by these stressors may not be severe enough (e.g. in comparison with the N3 treatment) to affect this maturation process. These results highlight the sensitivity and complexity of the mechanisms underlying the actions of TH on sensory system maturation, offering an avenue for future research.”

19. Line 108 – change nostril to olfactory organ, and change lateral line to trunk canal and correct this throughout.

R: Changed as requested.

20. Line 114 – what study looked at olfactory and visual cues, and is it known that those cues were olfactory and not gustatory. If this is not known please use “chemical cue” or “chemical choice experiment” and not “olfactory cue” or “olfactory choice experiment.” The same arguments can be used for the term “olfactory” on line 432 unless the cited papers have clearly demonstrated that the behavior is guided by olfaction and not gustation.

R: Good point. We now use “chemical” instead of “olfactory” as suggested.

21. Line 205 & 210 – “anthropogenised” is not a word. Please change this.

R: Changed as requested, we now use another wording (lines 246-255):

“Acute exposure to increased temperatures of +1.5 and +3°C or CPF levels spanning from 1 to 30 $\mu\text{g L}^{-1}$ therefore reflects the temperature fluctuations^{49,50} and potential or future pesticide fluctuations⁵² that larval fishes may experience when recruiting to coastal nurseries under high anthropogenic influence. Larger and older fish are less vulnerable to TH signaling disruption and associated neurological defects than metamorphosing fishes⁵⁶, indicating that exposure to acute stressors may alter fish predator-prey dynamics during this critical temporal window. As small declines in survival during recruitment can have dramatic consequences for population replenishment, our results raise concerns about the future of coastal fish nurseries, which, under the threats of climate change and anthropogenic stressors, could turn into ecological traps⁵⁷.”

22. Line 214 – the argument for the fitness consequences needs better support with citations.

R: This sentence was intended to summarize our results and therefore does not require external references. However, we can see that the term fitness was probably not appropriate and we have therefore changed our phrasing (lines 257-259):

“Overall, this study highlights that short-term exposure to acute anthropogenic stressors has detrimental consequences for the development and survival of metamorphosing fish by affecting their TH endocrine function.”

23. Line 218 – please define “replenishment potential”.

R: To make our point clearer, we have changed this term to “resilience” (line 261).

24. Line 247 – please more precisely define “settlement stage”. What was the condition of the other markers of metamorphosis such as meristic counts, presence of scales, and transitioning or juvenile pigmentation?

R: As mentioned in our reply to comment #3, settlement refers to the period when larval fish move from the ocean to the reef (i.e. reef entry or reef colonization). Our study species, *A. triostegus*, has a transparent body at settlement, and dark horizontal bars appear on the body flanks ~4 hours after settlement (Holzer et al., 2017). Transparency is therefore a reliable criterion to attribute to settlement-stage in this species (Holzer et al. 2017). We have added more detail to our Methods section to help any confusion by other readers (lines 299-301):

“Settlement-stage *A. triostegus* (i.e. fully transparent individuals¹¹, here define as day 0 (d0) individuals) were collected on the north-east coast of the island (S17°29'49.7362", W149°45'13.899") at night using a crest net¹¹ as they transitioned from the ocean to the reef.”

Please see our reply to comment #18 for discussion about the presence of scales.

25. Line 335 – the use of the term histology is correct here, but because tissue sectioning (the more common definition of histology) is not involved please remove the term histology from lines 371 and 398.

R: Agreed and changed accordingly.

26. Line 401 – There are newer reviews of lateral line structure and function that should be cited here. For instance the 2014 volume called “the lateral line system” (Springer).

R: Thank you for this suggestion. This citation has been added.

Reviewer #2

General comment:

*In this manuscript, Besson and colleagues demonstrate that disrupted TH signaling leads to aberrant development of sensory structures in *A. triostegus*, and that blocking TH signaling impairs the ability of fish to avoid predators. The authors then show that increased temperatures or CPF exposure during metamorphosis lower T3 and T4 levels, impair development of sensory structures and decrease survival in a predation assay. Survival in the predation assay is rescued by treatment with T3, and the authors conclude that TH disruption is the causal mechanism underlying decreased survival in fish exposed to higher temperatures or CPF. Additional data would strengthen this causal link, and the experiment needs an additional control (CT injected with T3). Especially without data showing that the T3 treatment rescues sensory structure development or avoidance behavior, the authors should explore alternative explanations for their increase in survival in T3-rescued fish. However, overall, these findings are novel, timely and important, the manuscript is well written, and the data are presented beautifully. Sarah McMenamin*

R: We would like to thank Dr. McMenamin for the positive and insightful review. We have addressed all comments, which we believe have strengthened the paper substantially.

Specific comments:

1. Some citations are repeated (9 and 53).

R: Thank you for pointing this out. This has been fixed.

2. Line 81 should include the citation Hu et al 2019 Dev Dyn, which shows that TH is essential for development of the lateral line system in zebrafish. The lateral line results should be compared to the results in Hu et al.

R: Thank you for bringing this study to our attention. It is now cited multiple times (e.g. lines 87, 184, and 190).

3. The introduction and discussion would be strengthened by mentioning that aquatic organisms are even less buffered against climate change than terrestrial organisms, citing work including Pinsky et al 2019.

R: We have added this reference and made the suggested changes, e.g. in the Introduction (lines 57-59):

“The negative impacts of anthropogenic stressors can be felt by all organisms at all life stages; however, aquatic species may be more vulnerable than terrestrial organisms⁷.”

Again, in the Discussion (lines 231-233):

“Given that stressors are rarely experienced individually, these results highlight the vulnerability of aquatic organism endocrine functions and the risk posed by anthropogenic stressors^{7,9}.”

4. In Fig 2, it is unclear to me why there seems to be a difference in the time spent at stimulus area with no cue between the control and T3 treated fish? Is this apparent difference significant and can it be addressed?

R: This difference is intriguing, indeed, but not significant ($t = 1.982$, $df = 17.14$, $P = 0.064$). We observed a lot of variability in our flume experiments (e.g. see how the data points range from 0 to 100 on Fig. 2a) and this difference could therefore just be a product of this variation. That being said, this comment raises a good point and we have removed the term “higher” when referring to the chemical preferences from the T3-treated fish. The text now reads (lines 123-126):

“In chemical choice experiments, d2 CT fish showed a clear avoidance of predator-cues (Fig. 2a). This response was similar in T3-treated fish, while N3-treated fish did not discriminate between water sources, similar to d0 fish (Fig. 2a).”

5. In lines 183-18, the authors say that temperature has a higher impact on T4 levels than CPF, and this is a bit unclear to me. It looks in the figure like the +3°C treatment results in a similar T4 level as CPF30. Do the authors mean that the ratio of T4/T3 is different between the different treatments? My eye is not detecting ratio differences either. Can this be clarified? If the goal is to highlight differences in T3/T4 ratios, it would be very helpful to show a graph with T3/T4 ratios for each individual in each treatment.

R: This is a good point, thank you for bringing it up. Initially we only looked at the effects of temperature and CPF on T4 and T3 levels, but not on the T3/T4 ratio. To address this, we have now added an additional analysis and added a supplementary figure (Supplementary Fig. 2). These results show that, in addition to CPF30 and +3.0°C exposures decreasing both T4 and T3 levels (Fig. 3a,b), CPF30 exposed fish also experienced a significantly lower T3/T4 ratio than +3.0°C exposed fish (Supplementary Fig. 2). This adds support to our previously presented results and more correctly shows that temperature had a greater effect on T4, and CPF on T3. Specifically, we have added the following to the Results (lines 149-151):

“Exposures to +3°C or CPF30 were therefore associated with comparably impaired maturation of sensory organs and TH disruption, but CPF30 fish experienced a lower T₃/T₄ ratio than +3.0°C fish (Supplementary Fig. 2).”

We have also added the following in the Discussion section (lines 222-227):

“Temperature had a higher impact on T₄-levels than CPF (Fig. 3a), as indicated by a higher T₃/T₄ ratio (Supplementary Fig. 2). This suggests a greater central effect, most probably at the neuroendocrine level, which may alter thyroid activity and thus explain the decreased levels of T₃ that we observed⁴⁵. In contrast, CPF had a greater effect on T₃ levels than temperature (Fig. 3b), as indicated by a lower T₃/T₄ ratio (Supplementary Fig. 2), suggesting a more downstream or peripheral effect, possibly on T₃ metabolism^{46,47}.”

6. Lines 190-192, the authors say that exposure to both +3°C and CPF5 or CPF30 causes death. I do not see these data presented in the manuscript, and the citation listed (Wu et al 2017) does not contain that experiment. This needs to be clarified.

We agree and we have removed this sentence. The text now reads (lines 233-237):

“Given that stressors are rarely experienced individually, these results highlight the vulnerability of aquatic organism endocrine functions and the risk posed by anthropogenic stressors^{7,9}.”

7. The authors show in Fig 3F that the survival of fish exposed to high temperatures or to CPF can be rescued by T3 injections. This is a very exciting finding, but it left me wondering about the actual behavior of these rescued fish and the development of their sensory organs. Is it possible to show data for the rescued fish in some or all graphs in Fig 1, 2 and Fig 3 C-D? Without showing these data, it is difficult to establish a mechanistic link between the T3 rescue and the increase in survival. Further, are there data for a T3 treatment on control fish? Do these show any difference in survivorship from euthyroid controls? (Also, do they show differences in development of sensory organs or predator avoidance?) This control is essential in interpreting the findings. My lab has found that hyperthyroid fish are generally hyperactive and tend to swim very fast and erratically (this is an unpublished observation). I wonder if the rescue you see in survival rates is due to the hyperthyroid fish swimming faster and more erratically, rather than actually having properly developed sensory structures and rescued appropriate predator avoidance. Alternatively, is it possible that the T3 treated fish smell or taste different to the predators, and could that explain the decrease in predation? These possibilities should be explicitly tested or at least addressed as possibilities.

R: We thank Dr McMenammin for this valuable point. As there are many relevant aspects in this comment, and we want to make sure that they are addressed properly, we would first like to provide context for our response. The work that we conducted in this study was hypothesis-driven, and the successive experiments that we conducted provided support for our main hypothesis – anthropogenic stressor induced survival defects in juvenile coral reef fishes are a result of TH disruption. The logic behind the successive components of this study were as follows:

1) We investigated whether, and showed that, TH signaling regulates sensory organ maturation (Fig 1e-g) and affects anti-predator behaviours (e.g. diminished ability to perform in a chemical choice experiment, diminished ability to perform in a visual preference experiment (Fig. 2a,b) and diminished ability to perform in a predation experiment (Fig. 2c)).

2) Based off these results, we then investigated whether, and showed that, temperature and CPF comparably disrupt T3 and T4 levels (Fig 3a,b), caused similar developmental defects of their sensory systems (Fig. 3c-e) and comparably decreased survival (Fig. 3f). We consider these results key to this study, as they show that anthropogenic stressors and pharmacological disruption of TH processes are associated with comparable sensory maturation defects, and that disruption of TH is associated with diminished anti-predator behaviors and a decreased ability of recruiting fish to avoid predation.

To examine the causality of these results, we hypothesized that supplementing exposed fishes with T3 would reverse the negative effects of TH disruption. Given that we worked with a wild species and that our work required collecting larval fishes *in situ* as they settled to the reef at night, we were inherently limited in the number of individuals that we could collect, and thus needed to prioritize which experiments were most valuable to examine our hypothesis. We therefore decided that, based on the results of our preceding experiments, that examining whether the survival defects that we observed, which were associated with TH disruption, were reversed when supplemental T3 was administered was the most important test. Indeed, it provides support for our hypothesis and demonstrates the survival consequences of TH disruption, and survival is the ultimate ecological output during recruitment. Our results are consistent with this, and we therefore believed that it was reasonable to conclude that our results support the hypothesis that anthropogenic stressor-induced TH disruption is a causal mechanism underlying decreasing sensory system maturation, diminishing ecologically important behaviors and decreasing survival prospects.

This being said, we agree with Dr McMenammin that this point is important, and have taken the following steps to address it:

A) Our results already demonstrate that T₃ injections promote sensory organ maturation and predator avoidance in comparison to control fish (Figs 1, 2), that TH disruption in stressor exposed fish is correlated with impaired sensory organ maturation (see Fig. 3a-e) and that stressor exposed fish that receive supplemental T₃ recover predator avoidance abilities in a predation experiment (Fig. 3f). To increase our evidence in support of supplemental T₃ increasing sensory system maturation, we now provide additional data (Supplementary Fig. 1, which we also include in this reply (figure on the right)) that we had performed on fish that were treated with both N3 and T3 treatments. This experiment was conducted to test whether T₃ and NH₃ compete (following Holzer et al. 2017 – eLife 6:e27595), but was not included in the previous version of the manuscript. In this experiment, we compared the number of lamellae at d2 in CT, T3, N3 and N3T3 treated fish, and found that the maturation of the sensory system was rescued in the N3T3 individuals. While these data only concern the nostril lamellae, it does provide further support in favor of supplemental T₃ rescuing sensory organ maturation in TH disrupted fish. To this end, we have added the following to the Results (lines 116-117):

“Supplemental T₃ rescued olfactory organ maturation in N3-treated fish (Supplementary Fig. 1).”

We also added the following to the Discussion (lines 191-196):

“Despite this advanced stage of development, we found that fish with pharmacologically promoted TH signaling experienced faster sensory organ maturation, those with disrupted TH signaling (either pharmacologically (Fig. 1e-g) or environmentally (Fig. 3c,e)) experienced impaired sensory development, and those with pharmacologically disrupted TH that received supplemental T₃ experienced rescued maturation of their olfactory organ (Supplementary Fig. 1).”

B) While our above response provides further support that sensory maturation is rescued when T₃ is administered, Dr McMenemy’s comment does highlight that the manuscript would benefit from a more cautious approach in the language used. Therefore, following her suggestion, we have adjusted the text and highlighted that further investigation into the effects of TH on sensory system maturation should be an important focus for future research. Specifically, we have added the following to the lines 203-204:

“These results highlight the sensitivity and complexity of the mechanisms underlying the actions of TH on sensory system maturation, offering an avenue for future research.”

Further, we have removed the words “the causal mechanism” and have replaced it with “a mechanism” (line 178).

In the abstract, we have also removed “Both stressors decreased fish thyroid-hormone levels, causally impairing sensory development and increasing predation vulnerability” and changed it as follows (making special note to remove the word “causally”) (lines 41-45):

“We then show that high doses of a physical stressor (increased temperature of +3°C) and a chemical stressor (the pesticide chlorpyrifos at 30 µg L⁻¹) induced similar defects by decreasing fish TH levels and impairing their sensory development. Stressor-exposed fish experienced higher predation; however, their ability to avoid predation improved when they received supplemental TH.”

As suggested by the reviewer, we agree that other processes may also contribute to rescuing the survival capacity of stressor-exposed / T₃-treated fish. We have therefore included the potential reasons brought by the reviewer in the discussion of our revised manuscript (lines 211-218):

“While we found that T₃-treated fish exhibited more rapid sensory organ maturation (Fig. 1e-f) and comparable responses to predator cues to control fish (Fig. 2a-b), which is consistent with an accelerated sensory development facilitating more effective anti-predator abilities, we cannot rule out the possibility that supplemental T₃ may have also impacted their behavior in other ways. For example, supplemental T₃ may affect swimming behavior (e.g. as suspected in *Danio rerio*, McMEnamin pers. comm.) and might alter the cues emitted by these individuals (e.g. smelling or tasting different to the predators), which may have contributed to their ability to avoid predation (Fig. 2c).”

Reviewer #3:

General comment:

*In this manuscript, the authors aim to investigate the physiological mechanism underlying observed effects of anthropogenic stressors on fish behaviour and survival. They tested two stressors: increased temperature and a chemical stressor (chlorpyrifos pesticide), and investigated the effect of these on the metamorphosis stage using the convict surgeonfish (*Acanthurus triostegus*). Metamorphosis is controlled by thyroid-hormones (TH), wherefore the effects of these stressors on TH levels was investigated. They also investigated the effect of TH levels and the two stressors on predator avoidance behaviour and survival. The authors report that both stressors decreased TH levels and increased predation vulnerability. The manuscript is impressive in terms of containing a lot of data. The experimental design is thorough, with the authors first investigating sensory organ maturation and the role of TH during metamorphosis (with the TH signal disruption treatment, N3, showing repressed sensory organ maturation), then the effect of TH on behaviour (with N3 treated fish showing no response to chemical or visual cues from a predator, which led to lower survival). These experiments were followed by investigations of the effects of the stressors (separately and combined) on TH signalling and sensory development, and whether TH injections can restore survival rates. Several treatment levels were used, in combination with controls and solvent controls, as well as some experimental sampling during different days of the metamorphosis. This also means that there are many different treatments to keep track of for the reader. The authors did a pretty good job at presenting the data in a clear and concise manner, but due to the high number of treatments it is still sometimes difficult to follow all results. The experimental design is quite well described, and I commend the authors for being clear, for example on number of replicates removed and which criteria that were used for removing those (e.g. page 18-19, line 544-464, line 480-482)).*

R: We thank the reviewer for their positive appraisal of our manuscript. The comments provided were very helpful, and we believe that they have increased the overall quality of the manuscript.

Specific comments:

1. However, several important method aspects are missing, such as which year each experiment was performed (coupled with a lack of testing for year effect in the statistical analyses)?

R: This is a good point. We have provided this information in the revised manuscript in a dedicated Method section entitled “Study period and site” (line 293). In this section we state (lines 288-296):

“This study was conducted from February 2015 to June 2018, at Moorea Island, French Polynesia (S17°32'16.4589", W149°49'48.3018"). Sampling for the examination of sensory organ maturation under pharmacological treatments were conducted in 2015. Sampling for the investigation of behavioral preferences and survival to predation under pharmacological treatments, as well as sensory organ maturation under anthropogenic stressor exposure were conducted in 2016. Sampling for the examination of survival to predation under anthropogenic stressors exposure, and TH levels under co-exposure to anthropogenic stressors were conducted in 2018. Sampling for the investigation of TH levels under single anthropogenic stressors were conducted in both 2016 and 2018.

Regarding the question about whether experiments were conducted across multiple years, we appreciate that this would not have been clear in the previous version of the manuscript, but note that only one experiment occurred across multiple (i.e. two) years and that this was considered in our statistical analyses (as a random effect). This was initially not clearly indicated in our Methods, but is fixed now (lines 587-592):

“Gamma generalized linear mixed effect models (GLMEM) were used to assess if anthropogenic stressor exposures influenced TH levels and T₃/T₄ ratios⁷⁸. TH level or T₃/T₄ ratios were used as the dependent variable, and replicate was included as a random factor to account for differences in TH levels only due to the two different *Cobas* analysers that were used in the two different years, as well as for potential differences between years.”

Further, we investigated T₃ levels in d0 *Acanthurus triostegus* across two seasons and three lunar phases in 2015. This work reveals that there is no variation in T₃ levels across these seasons and lunar phases and this is why we did not take these parameters into account in this study. We thought this would be relevant to mention this following the reviewer’s remark. Here is the figure associated with this result (see below), which has been added as a supplementary figure (Supplementary Fig. 14). This result is now brought in our Method section (lines 592-594):

“As preliminary experiments provided no evidence that season and lunar phase affected T₃ levels in metamorphosing *A. triostegus*, we did not include them in our analyses”

2. *How many in situ cages and aquaria that were used for housing and exposing the fish (coupled with a lack of tank-effect analysis)?*

R: We agree that more explanation would be beneficial here. Randomizing prior to experimentation is a design feature that exists to mitigate a biasing effect (i.e. a tank effect in this instance) before it occurs, while testing afterwards is important if there is a reasonable reason to suspect that a biasing effect might exist. For example, if our cages/tanks were fixed in place and experienced different conditions then it certainly would have been appropriate to test for a tank effect. However, this was not the case for us (i.e. the cages/tanks were all identical and were regularly set up and packed up according to when we had fish available to test). We were very conscious of using the most elegant experimental design possible for this study, and therefore incorporated the randomized use of tanks into our design. This approach meant that there was no reasonable reason to expect, and therefore test for, a tank effect. This being said, we have highlighted in the text that 12 in situ cages and 12 aquaria were used throughout the study and that use of any given cage/tank was randomized prior to each trial (lines 333-334 and 340-342).

3. *Method description and data for how the warming and chemical treatments were maintained and measured. I did not find any description or data of analysis of water and fish samples from the chemical treatment to verify that the targeted levels were reached?*

R: Concerning the exposure to increased temperatures, we have added supplementing information in our Methods (lines 350-356):

“Heaters controlled by thermostats were used to control the temperatures in the increased temperature experiment. Before the experiment, each tank was in open circuit with temperature maintained at 28.5°C. At the beginning of the experiment, fish were introduced in the aquarium and the thermostat temperature was then set up to 28.5°C, 30.0°C or 31.5°C according to the treatment. The temperature of interest was reached within 2 hours. Temperature in each tank was then visually checked (on the thermostat controller) at least 5 times per day and never differed from the target temperature by more than 0.2°C.”

Concerning the CPF exposure, the CPF concentrations that are indicated in the manuscript are nominal concentrations. We did not measure CPF concentrations in the water, but a previous study (Botté et al. 2012 - Mar. Pollut. Bull. 65, 384–393) using similar methods measured 3.53, 13.9 and 42.7 $\mu\text{g L}^{-1}$ of CPF in seawater 24 hours after spiking nominal concentrations of 4, 16 and 64 $\mu\text{g L}^{-1}$. Authors of this study consequently

estimated that approximately 80% of nominal concentrations were measured after 24 hours, and we therefore expected a similar stability in our study. We have edited our Methods to more make this clearer (lines 360-374):

“For CPF exposure, five different treatments were applied: unaltered seawater (CT, control treatment), seawater with acetone at a final concentration of 1:1.000.000 (CPF0, solvent control treatment, as CPF was made soluble using acetone), or seawater with CPF at a nominal concentration of either 1, 5, or 30 $\mu\text{g L}^{-1}$ (CPF1, CPF5, and CPF30 treatments), based on the findings of recent studies of reef fishes exposed to CPF^{11,14,65}. CPF was spiked in each tank from dilutions that were prepared in advance: 1 $\mu\text{g L}^{-1}$, 5 $\mu\text{g L}^{-1}$, and 30 $\mu\text{g L}^{-1}$. From these dilutions, 12 μL were pipetted and spiked in the 12 L exposure tanks, therefore reaching nominal concentrations of 1 $\mu\text{g L}^{-1}$, 5 $\mu\text{g L}^{-1}$, and 30 $\mu\text{g L}^{-1}$. Similarly, 12 μL of acetone was spiked in the tank for the CPF0 condition. Spike was allowed to mix for 2 minutes (water mixing due to the air stone) before fish were introduced in the tank. At the end of the 32-hour exposure, CPF concentrations in the water or in the fish tissues were not evaluated as we were only interested in the effects of CPF spikes on fish metamorphic processes. Nevertheless, a previous study using similar methods and nominal concentrations of similar magnitude (i.e. ranging from 4 to 64 $\mu\text{g L}^{-1}$) measured CPF levels corresponding to 80% of nominal concentrations after 24 hours⁶⁶, therefore suggesting a good stability of CPF levels in the condition of our study.”

4. Also, the statistical methods used are rather simple, and while complicated statistical tools should not be a target in itself, more modern tools could be more appropriate, for example rather than using non-parametric tests, which was used in the manuscript for certain analyses.

R: We have conducted completely new statistical analyses based on more modern tools to comply with this remark. We have edited our statistical method paragraph, which now reads (lines 580-599):

“All statistical analyses were conducted using *R version 3.5.3*⁷⁶. Conway-Maxwell-Poisson (COM-Poisson) generalized linear models (GLM) were used to assess if pharmacological and anthropogenic stressor treatments influenced the number of lamellae, the number of trunk canal pores, and the number of survivors to the predation experiment⁷⁷. COM-Poisson GLM were also used to assess if pharmacological treatments influenced the number of pes. Linear models (LM) were used to assess if pharmacological treatments influenced the bpc, ggc and prn densities, and if pharmacological and anthropogenic stressor treatments the Fulton’s K condition factor. LM were also used to assess if the trunk canal length varied across metamorphosis. Gamma generalized linear mixed effect models (GLMEM) were used to assess if anthropogenic stressor exposures influenced TH levels and T_3/T_4 ratios⁷⁸. TH level or T_3/T_4 ratios were used as the dependent variable, and replicate was included as a random factor to account for differences in TH levels only due to the two different *Cobas* analysers that

were used in the two different years, as well as for potential differences between years. As preliminary experiments provided no evidence that season and lunar phase affected T_3 levels in metamorphosing *A. triostegus*, we did not include them in our analyses (Gamma GLMEM, Supplementary Fig. 14). For each model, diagnostic plots were examined and outputs compared to raw data to confirm goodness-of-fit and residual homoscedasticity, and, when applicable, residual normality was assessed using Shapiro-Wilk Normality Test. Paired t-tests or Wilcoxon signed rank tests were used to assess whether fish spent more time in the no cue choice area vs predator cue choice area, depending on residual normality (Shapiro-Wilk Normality Test)”

5. Another general comment is the overall emphasis on the negative effects found. In particular in the title and abstract, but in general the paper gives the impression that these stressors had a negative impact on sensory development and behaviour, period. However most of the effects were seen in the most extreme treatments, i.e. +3°C and the CPF 30 µg/L treatment (e.g. as shown in Fig. 3), while the lower temperature (+1.5°C) and the lower levels of CPF (1 and 5 µg/L) only had effects on some of the measurements, effects that were not consistent. The fact that these treatments had no effect is mentioned in the result section, but largely ignored in the rest of the manuscript. The authors need to make it clear that the lower treatments had none, or a smaller and non-consistent effect.

R: Agreed. We have placed greater emphasis on the fact that we observed effects only for high doses of anthropogenic stressors, and that lower exposure levels had apparently none, smaller and non-consistent effects. First, in the Abstract (lines 41-43):

“We then show that high doses of a physical stressor (increased temperature of +3°C) and a chemical stressor (the pesticide chlorpyrifos at 30 µg L⁻¹) induced similar defects by decreasing fish TH levels and impairing their sensory development.”

Then, in the Discussion (lines 220-230):

“While exposure to high levels of increased temperature and CPF both inhibit sensory development and reduce a fish’s likelihood of avoiding predation, the two stressors affected endocrine signaling differently. Temperature had a higher impact on T_4 -levels than CPF (Fig. 3a), as indicated by a higher T_3/T_4 ratio (Supplementary Fig. 2). This suggests a greater central effect, most probably at the neuroendocrine level, which may alter thyroid activity and thus explain the decreased levels of T_3 that we observed⁴⁵. In contrast, CPF had a greater effect on T_3 levels than temperature (Fig. 3b), as indicated by a lower T_3/T_4 ratio (Supplementary Fig. 2), suggesting a more downstream or peripheral effect, possibly on T_3 metabolism^{46,47}. This variety of action modes may explain the synergistic effect of both

stressors on TH levels with co-exposure to 1.5°C or CPF5 causing TH disruption (Fig. 4a-b), while these stressor levels did not affect TH levels when exposed separately (Fig. 3a,b).”

6. I also have some concerns regarding the choice of the targeted chemical levels. The chlorpyrifos (CPF) levels used were 1, 5, and 30 µg/L. These levels are argued to be biologically relevant in the discussion (page 8, line 197-199). However, the cited report (reference 45, NRA Review of Chlorpyrifos) describes that chlorpyrifos is an occasional contaminant of surface waters, and that levels are usually below 1 µg/L. These values are from rivers, which would be relevant if tested on a freshwater organism inhabiting such areas. Since this manuscript use a marine species, levels measured in the sea where they occur are needed in order to say what is biologically relevant. If levels are usually below 1 µg/L in rivers, then those values would be much lower in the oceans due to the dilution effect. The report state that only a few high outliers have been measured, where levels reached 25-26 µg/L, this was again in rivers and irrigation drainage. Based on this, I would say that even the lowest level used here is not biologically relevant, and the highest level used is even higher than extreme outliers. The results are still interesting, knowing the effects of high levels of contaminants can be important, for example it is highly likely that levels will be higher in the future. However, it should not be called biologically and/or environmentally relevant. Regardless the levels used here, any values of this contaminant measured closer to where the fish were collected, or at least in a coral reef environment, would be much more relevant to cite. The other references used by the authors to justify the levels used are for example reference 46 (Bigot et al. 2016), however they report levels of 0.18-0.54 pg/L, i.e. magnitudes lower than what was used here. Reference 57 (John & Shaike 2015) does not seem to include much data on environmentally relevant levels, but for example discuss toxicity and LC50 values. Reference 13 (Besson et al. 2017) do state that Australian reef surface waters can reach up to 1 µg/L, but I was unable to find the paper that is cited for this value (NRA. in National Registration Authority for Agricultural and veterinary Chemicals 1, 17 2000), meaning that the validity of this data point cannot be checked or confirmed.

R: We agree that the discussion of these concentrations was not appropriate. We have substantially modified this discussion paragraph to comply with this remark (lines 238-249):

“Under these circumstances, acute fluctuations such as temperature spikes of +1.5°C and +3.0°C are environmentally relevant, and CPF levels from 1 to 30 µg L⁻¹ are informative in the context of decreasing water quality in coastal areas in response to increasing pesticide use and land clearing⁵¹. Indeed, rapid temperature shifts are common in coastal surface waters and can reach up to 12°C following solar and tidal forcing⁵⁰. Likewise, while CPF levels in contaminated surface waters are generally below 1 µg L⁻¹ with limited persistence in the water column, these levels can spike up to 26 µg L⁻¹ in rivers⁵². This suggests that following run-offs, and on a short time scale such as the 32-hour exposure of our study, CPF levels in coastal waters could largely exceed the pg/ng per liter

concentrations usually reported in seawater⁵³⁻⁵⁵. Acute exposure to increased temperatures of +1.5 and +3°C or CPF levels spanning from 1 to 30 µg L⁻¹ therefore reflects the temperature fluctuations^{49,50} and potential or future pesticide fluctuations⁵² that larval fishes may experience when recruiting to coastal nurseries under high anthropogenic influence.”

We have also removed the John & Shaike 2015 - Environ. Chem. Lett. 13:269–291, and Besson et al., 2017 Sci. Rep. 7:1–9, references from this paragraph, and we have added two new references (Chernyak et al., 1996 - Mar. Pollut. Bull. 32:410–419; and Zhen et al., 2019 Environ. Pollut. 252: 573–579), which are more appropriate.

7. In relation to this, it could also be discussed how biologically relevant a 36h (page 13, line 308-309) chemical exposure period is?

R: It is not a matter of duration but rather a matter of sensitivity and vulnerability of this specific temporal window. As indicated in our reply to the previous remark, we have discussed how this acute exposure is relevant in the context of short-term environmental fluctuations. In the results, we emphasize how relevant this short term exposure is in term of the ecology of reef fishes at recruitment (line 120):

“As predation is high in the two days following settlement in reef-fishes¹⁶”

This exposure duration is also developmentally relevant, as most metamorphic processes occur within two days post-settlement (see Holzer et al., 2017 – eLife 6:e27595, which we refer to and cite numerous times within our current study). We would like to emphasize that endocrine disruption is common on such short temporal windows: see Newbold 2004 - Toxicol. Appl. Pharmacol. 199: 142– 150; Parsons et al., 2019 Aquat. Toxicol. – 209:99-112)

8. Overall, I think the manuscript has good potential to be of high value to many fields, but these concerns must be clarified first. Specific comments: Was any data collected blind regarding treatment? I could not find such a statement so I assume it was not. This is fine, sometimes blind data collection is impossible, however this should be stated in the paper. In particular given the choice flume experiment was not video recorded despite only lasting rather short period of time (14 min for the choice flume experiment, out of which 10 min were used for data collection), and the apparent availability of a camera, which was used in the second behavioural experiment testing visual cues (page 19, line 477-478).

R: We agree that we could have been clearer here. The data were not collected blind, which is stated in our NR-reporting-summary. We would like to highlight, however, that the choice flume experiment was video recorded. We have modified our Methods to state this more clearly (lines 523-526):

“After releasing the fish in the choice arena, a 2 min acclimation period was observed, then fish position (left or right choice area, or drain area) was recorded every 2 sec for 5 min (Supplementary Fig. 10), using a camera (GoPro Hero 2) located above the edge of the flume tank.”

9. Throughout the manuscript, please change the word “olfaction”, unless you are certain that this is the sensory system under investigation (which most of the time you are not, since the other sensory systems (chemoreception and taste) were not blocked). Use the word “chemosensory cues”, or similar.

R: We have removed the word “olfaction” throughout the manuscript (as also requested by reviewer 1). We now use the word “chemical” to refer to the chemical cues presented in the flume experiment, which we now refer to as “chemical choice experiment”.

10. Page 10, line 246: The study was conducted over several years (2015-2018). Which experiment was conducted when? Could there be differences between years? This was not statistically tested.

R: Please see our reply to comment #1, which addresses this issue.

11. Page 11, line 249-252: How many cages were used for the thyroid hormone signaling experiment? On line 277 it says 15 fish were kept in each in situ cage, but not how many cages that were used. Similarly, for the temperature and chlorpyrifos exposure experiments, how many aquaria were used? This also needs to be incorporated in the statistical analyses.

R: Please see our response to comment #2, which addresses this issue.

12. Page 12, line 290-296: Please describe the method used to increase temperature, how it was maintained and controlled. Please include data on actual measured temperature per tank and treatment. How was the increase applied? I.e., using some ramping protocol? Or were the fish put straight from 28.5°C into the warm treatments?

R: Please see our reply to remark #3, which answers to the same concern.

13. Page 13, line 300-303: Please add methods describing how the chemical treatments were obtained (e.g., was the chemical simply mixed into each tank together with solvent?). In particular, give methods for how the target concentrations were measured and controlled. Please include data on actual measured concentrations per tank and treatment. In ecotoxicology it is standard that not only the treatment water concentrations are analysed, but also fish tissue. This was not done here (and the way the methods are written, it is not even given that water concentrations were measured). This means that the actual concentrations in the fish are unknown, and also so in the water?

R: The actual concentrations in fish and in water were, as already mentioned, not measured, but we do not consider this an issue as we were not interested in fish bioaccumulation of CPF but rather to the effects of acute exposure to waterborne CPF onto fish metamorphic process. Also, a previous study demonstrated good stability of CPF in conditions similar to our study (Botté et al. 2012 - Mar. Pollut. Bull. 65, 384–393). Please see our reply to remark #3 from this same reviewer for a detailed answer regarding this. We have also added in our Methods section additional information regarding how CPF spikes were performed (lines 364-369):

“CPF was spiked in each tank from dilutions that were prepared in advance: 1 $\mu\text{g } \mu\text{L}^{-1}$, 5 $\mu\text{g } \mu\text{L}^{-1}$, and 30 $\mu\text{g } \mu\text{L}^{-1}$. From these dilutions, 12 μL were pipetted and spiked in the 12 L exposure tanks, therefore reaching nominal concentrations of 1 $\mu\text{g } \text{L}^{-1}$, 5 $\mu\text{g } \text{L}^{-1}$, and 30 $\mu\text{g } \text{L}^{-1}$. Similarly, 12 μL of acetone was spiked in the tank for the CPF0 condition. Spike was allowed to mix for 2 minutes (water mixing due to the air stone) before fish were introduced in the tank.”

14. Page 17, line 434-437: Please add the water volume in which the five predators were soaked for 2 h to create the predator odour. The concentration of predator cue cannot be estimated based on the given information (meaning the experiment could not be repeated for this particular point).

R: This information was now added in the Methods (lines 514-515):

“five *Lutjanus fulvus* predators were soaked, into a 125 L tank, for two hours prior to the experiment”

15. Page 18, line 454-455: Are there any implications of testing a diurnal species at night? Red light was used, which is good, but the fish would not be expected to be active during night? In particular given they are sensitive to light pollution (artificial light at night, as stated on page 10, line 236-238). This should at least be discussed.

R: We agree that we could have been clearer here. Flume experiments were conducted at night only for d0 fish. The reason for this is that d0 fish actively settle to the reef at night (i.e. d0 fish are not diurnal), and this is therefore the most biologically relevant time to do this. We are not concerned about a light pollution issue, as the experiment was conducted under red light. We have now made this clearer in the text (lines 532-534):

“d0 fish were tested immediately after collection (i.e. at night) as this is when they are actively moving from the ocean to the reef, and is thus the most biologically relevant time to do so.”

16. Page 19, line 477-478: For the second behavioural experiment, testing visual cues, the experiment was video recorded (please add details on camera used, placement of camera etc., and how the videos were analysed, was some software used?) in order to limit disturbance by an observer. However, in the choice flume experiment, an observer was present (page 18, line 453). How did you ensure that the observer did not cause any disturbance in the first experiment, when this was a concern in the second behavioural experiment?

R: We agree that we could have been clearer here. We now provide the requested information about the camera that we used, how it was placed during the experiment, and how videos were analyzed (lines 555-558):

“Fish position (i.e. choice area 1, no choice area, choice area 2; Supplementary Fig. 11) was then assessed every two seconds over a 10 min, using a camera (GoPro Hero 2) to limit any external visual disturbances such as an observer’s presence. The camera was located above the choice tank.”

No software was used, as indicated in our NR-reporting-summary. We were not concerned about the observer causing visual disturbance during the flume experiment, as it was conducted in the dark under a red light. This was not the case for the visual choice experiment, which was conducted under lighted conditions, and therefore an observer was not present.

17. Page 20, line 492-493: In my experience, obtaining an ethical permit to perform experiment where a prey fish is to be consumed by a predator is very difficult. In particular if the experimental design means putting prey and predators together and counting the number of surviving prey after a certain amount of time, as done here. Usually such experiments, if allowed at all, require constant monitoring, so that harmed fish can be removed and euthanized to prevent hours of suffering. Please indicate clearly that the experimental design used here was specifically included in the ethical permits and approved, for example by enclosing the ethical permit where this design is mentioned (the ethical permits given on page 10, line 223-225 are not public documents and hence there is no possibility to review this).

R: We confirm that all experiments were conducted in accordance with the relevant governing bodies. The environment code of French Polynesia states that approval of - and permits for - scientific experiments are only

required for emblematic and protected species. For all other species that are not cited in the environment code of French Polynesia, and which are not listed on the CITES Trade Database, scientific experiments can proceed without approval/permit if the information gained increases understanding of coral reef function and contributes to short- and long-term protection of French Polynesian coral reefs. This is publicly available here: <https://www.service-public.pf/diren/partager/code/>. We have added this information in our Methods (lines 266-268):

“This study did not involve endangered or protected species and was carried out in accordance with the guidelines of the French Polynesia code for animal ethics and scientific research (<https://www.service-public.pf/diren/partager/code/>).”

18. Page 20, statistical analysis: The fish were housed in groups in aquaria or in situ tanks, but there is no mention of checking for tank effects? There's also no analysis of differences between years, and information on which experiment that was conducted what year is missing.

R: Please see our replies to remarks #1 and #2, which answer to the same concerns.

Reviewers' Comments:

Reviewer #1:

Remarks to the Author:

This manuscript looks at the effects of higher temperatures and/or the presence of a chemical stressor on the development of three sensory systems (olfactory organ, retina, lateral line canal on the trunk). As such it is a novel study and especially because it used a coral reef species - experimental work on the early life history stages of coral reef fishes are rare. Further, it looks at environmental stressors and their effects on morphological development, hormonal pathways and behavior - a nice integration! The statistics appear to be done properly, and data is presented carefully in graphic form. With respect to reproducibility, the major barrier would be the availability of study specimens, which for this study were collected in the field at a remote lab on an island in the Pacific.

The authors carefully considered the comments of the two reviewers of the first submission of this paper and made appropriate changes, which has improved the manuscript. Nevertheless, there are some issues that need to be resolved, with respect to word usage as they relate to the precise descriptions of the variables measured in the various experiments presented, and the presentation of the data in the figures.

The term "undermine" in the title is misleading for the same reason that the use of "impaired" (and "impairment"), "disrupt", "maturation" in other places in the manuscript. It is clear from the data that some aspects of the experimental treatments certainly "affected" the morphological development of the retina, olfactory organ (lamellae), and formation of pores of the lateral line canal on the trunk. And perhaps it is accurate to say that treatments affected the "timing" of the normal development of these structures. However, "maturation" suggests that there is some developmental endpoint (the mature condition), but that is not considered here since data on adults (sexually mature individuals) are not presented. Further, the structure-function relationship for density of bipolar cells in the retina, number of lamellae, and number of lateral line pores in the trunk canal are each unknown (although density of some retinal cells - the photoreceptors - not the bipolar cells, predicts visual acuity) - various studies using various species have tried to find functional correlates but failed. The changes observed in this study cannot necessarily be considered "impairments", or the "undermining" of development without knowing the functional correlates of these specific changes. Finally, while the retina and olfactory rosette are composed of sensory receptor cells, the number of lateral line canal pores reflects the development of the non-sensory accessory structures of the lateral line canal (the sensory organs, neuromasts, are within the canal; these are not studied here). Thus, the language used to describe the results of the experiments presented need to be modified so that they are not overstated; such changes in word usage will not change the outcomes of the experiments or the conclusions reached, but will make the report of the outcomes more precise.

That being said, behavioral responses to a visual predator stimulus, and to a chemical predator stimulus ("chemical" not "olfactory"; see caption in Supplemental Figure __) were indeed affected by some aspects of experimental treatments, implying that changes in the timing of sensory development did indeed affect behavior. In addition, overall survival (# of fish surviving; not "survival rate" [line ____]) was affected by experimental treatments. However, in this case, behavior was not specifically observed - although overall behavioral alteration must have resulted in a change in the ability of predators to capture prey [and/or for prey to avoid predators]). Again, the language used to describe the results of the experiments presented need to be modified so that they are not overstated; such changes in word usage will not change the outcomes of the experiments or the conclusions reached, but will make the report of the outcomes more precise.

One issue with behavioral experiment design and data analysis - fish that did not show a preference

are important – why were they deleted from the analysis (line 569)?

It should also be mentioned that the auditory system may be affected by the stressors.....It has been shown that larval fish respond to sound, so do not neglect this sensory system, which was not studied here.

More specific comments, for clarity:

Line 36/41 – please differentiate between metamorphosis and recruitment – these are two transitions, but “transition” (singular) is used in line 36.

Line 45/6 – suggestion – “Our results highlight the fact that two different anthropogenic stressors (physical, chemical) can alter critical developmental (morphological) and ecological transitions via the same hormonal pathway”

Line 60 – do you mean “metamorphosis” here?

Line 75-76 – I do not understand the logic in this sentence. Please check.

Line 85-87 – please delete “anthropogenic” – this is understood.

Line 87 – here and throughout – do not use TH because in different parts of the MS you are talking about one form and in other places you are talking about two forms. Just spell it out.

Line 106 – “rapid” – compared to what?

Line 108 – “surge” – compared to what? And is it # or density that was measured? Further, the # of pores is initially determined by the number of neuromasts in the canal (one pore on either side of a neuromast). So, is there a maximum number of pores reached during development?

Line 112 – “increased rate of maturation” – does a statistical test support this? And please see my comments about “maturation” above.

Line 112/113 – pores are not sensory organs – see my comment about this above.

Line 116 – did supplemental T3 rescue the retina or LL pores? If not, why not?

Line 122 – instead of CT in the text, can you spell out “control”? This would make the text easier to read.

Line 130-131 – “survival” – does this refer to the number of fish that survived, or the total % of fish that survived? Please clarify.

Line 143 – bpc – would it be possible to spell this out, or put in caps throughout, and in graphs? Every time I see it, it looks like a typo to me.

Line 154 – is it survival “rate” or the % of fish that survived? Check usage of “rate” throughout.

Line 164 – change “co-exposed” to “simultaneously exposed”

Line 165 – “suboptimal” – how was this determined?

Line 172 – “recruiting fishes” – perhaps say “settlement stage fishes”?

Line 176 – it is “presumed” anti-predator behaviors – behavior was not measured.

Line 178 – “apparently” recovered their ability to avoid....

Line 193 – “Promoted TH signaling”, should be “enhanced TH signaling...”

Line 196 – “rescued maturation _____ of their olfactory organ...” (but again, consider whether “maturation” should be used).

Line 199 – development (not maturation) of the trunk canal does include differentiation of canal neuromasts (not studied here) and enclosure and elaboration of the canal. The epithelial proliferation that occurs, that is relevant here, is the formation and elongation of the pores in the skin covering the lateral line canal.

Line 211 – “increased mortality” – OK, but refer to “higher numbers of fish captured by predators”, which is presumably due to a change in the developmental timing of the sensory organs.

Line 213 – 217 – please break into two sentences.

Line 261 – delete “detrimental” and add “changes behavior” at the end of the sentence.

Line 265 – say “populations and communities”.

Line 285 – say “metamorphosis is quickly followed by settlement and recruitment” (not “coincides”) –

this is a good place to reinforce that these are three different processes.

Line 362 – This sentence should be in the Intro to the paper, where the relevance of the stressors are laid out.

Line 409 – say “Retina”

Line 427 – please spell out “ds”

Line 429 – should be “another acanthurid species”

Line 432 – “processing efficiency of the retina toward visual cues” – please clarify. Perhaps “the ability of the retina to form an image”?

Line 439 – do you mean “thickness” instead of “width”?

Line 448 – say “number of folds or lamellae” – to define lamellae.

Line 450 – change “circulating” to “moving through each...”

Line 454 – shorten to “2.5% glutaraldehyde in 1M sucrose and 0.1M sodium cacodylate (pH 7.4)...”. Make same changes in line 456-7.

Line 476 – should be “the lateral line system and are ciliated sensory organs, composed of hair cells, like those in the inner ear, located...”

Line 479 – “at the level of the canal pore” is incorrect – not sure what is being described here. The neuromasts are within the short canal segments between pores.

Line 486 – what are “vertical plates” – are these the canal walls such that the roof of the canal is still just soft tissue?

Line 494 – “numbers” – yes, but please check throughout where “density” may have been used.

Line 519 – “odor” suggests olfaction. Can you just say “chemical cues from predator”?

Line 530 – arena or area – please check throughout and in figures and figure captions (main and supplementary).

Line 533 – Not sure what immobility has to do with anything about side preference. Please clarify and check throughout.

Line 537 – was this “in the dark with IR light” or was it “in the dark with red light”?

Line 552 – should be “in the presence of a visual predator”.

Line 594 – should be “and the Fulton’s K condition factor was used for pharm.....treatments.”

Line 595 – varied with age, not varied across metamorphosis; was d8 post-metamorphic?

Line 599 – should be “due to the fact that the analysis was carried out by two different people in two different years”.

Figures –

In ALL Graphs – please add the units of the variables on the Y-axis (e.g., # of lamellae, bpc density (#cells/____), # trunk canal pores), etc.

In all captions, results of statistical tests with P-values and F values must be added, esp. since these are not in the text. Then they can be eliminated from the figures themselves (use ** instead), which will make them less cluttered.

Figure 1 is quite information rich – Can the image be split from the graphs with the addition of the image in Supplementary Fig. 4a – for anatomical context. Similarly – can the image showing the nares in the Supplementary Figure be added here to accompany the rosette for context? These are likely not commonly understood since they are “fish-specific”. However, the cross section of the retina should be generally recognizable.

Suppl. Fig. 3 – “rosetta” should be “rosette” and add text to say how image b is related to image a (what was dissected away?). “SEM picture” should be “scanning electron micrograph”

Suppl. Fig. 4 – Fig. a should be moved to Fig. 1 – see comment above, then this figure can be eliminated, since b is in Fig. 1.

Suppl. Fig. 5 – change “ganglionar” to “ganglionic”, “layer width” should be “layer thickness” – see other comment above and check throughout.

Suppl. Fig 10 – should be “chemical” not “olfactory” in caption and figure.

Suppl. Fig. 12 – please add scale bar or mention tank size in caption.

Reviewer #2:

Remarks to the Author:

All concerns have been adequately addressed.

Response to Reviewer 1:

General comment: This manuscript looks at the effects of higher temperatures and/or the presence of a chemical stressor on the development of three sensory systems (olfactory organ, retina, lateral line canal on the trunk). As such it is a novel study and especially because it used a coral reef species - experimental work on the early life history stages of coral reef fishes are rare. Further, it looks at environmental stressors and their effects on morphological development, hormonal pathways and behavior - a nice integration! The statistics appear to be done properly, and data is presented carefully in graphic form. with respect to reproducibility, the major barrier would be the availability of study specimens, which for this study were collected in the field at a remote lab on an island in the Pacific. The authors carefully considered the comments of the two reviewers of the first submission of this paper and made appropriate changes, which has improved the manuscript. Nevertheless, there are some issues that need to be resolved, with respect to word usage as they relate to the precise descriptions of the variables measured in the various experiments presented, and the presentation of the data in the figures.

R: We would like to thank the reviewer for their thorough review, which greatly improved our manuscript. We have addressed all comments (see below).

Comment 1: The term “undermine” in the title is misleading for the same reason that the use of “impaired” (and “impairment”), “disrupt”, “maturation” in other places in the manuscript. It is clear from the data that some aspects of the experimental treatments certainly “affected” the morphological development of the retina, olfactory organ (lamellae), and formation of pores of

the lateral line canal on the trunk. And perhaps it is accurate to say that treatments affected the “timing” of the normal development of these structures. However, “maturation” suggests that there is some developmental endpoint (the mature condition), but that is not considered here since data on adults (sexually mature individuals) are not presented. Further, the structure-function relationship for density of bipolar cells in the retina, number of lamellae, and number of lateral line pores in the trunk canal are each unknown (although density of some retinal cells – the photoreceptors – not the bipolar cells, predicts visual acuity) – various studies using various species have tried to find functional correlates but failed. The changes observed in this study cannot necessarily be considered “impairments”, or the “undermining” of development without knowing the functional correlates of these specific changes. Finally, while the retina and olfactory rosette are composed of sensory receptor cells, the number of lateral line canal pores reflects the development of the non-sensory accessory structures of the lateral line canal (the sensory organs, neuromasts, are within the canal; these are not studied here). Thus, the language used to describe the results of the experiments presented need to be modified so that they are not overstated; such changes in word usage will not change the outcomes of the experiments or the conclusions reached, but will make the report of the outcomes more precise.

R: We have modified the language used in the manuscript to ensure that it is not misleading nor overstating. We have replaced “undermine” by “impact” in our title (line 2):

“Anthropogenic stressors impact fish sensory development and survival via thyroid disruption”

Similarly, we have replaced “impairing” by “affecting” in our abstract (line 43-46):

“We then show that high doses of a physical stressor (increased temperature of +3°C) and a chemical stressor (the pesticide chlorpyrifos at 30 µg L⁻¹) induced similar defects by decreasing fish TH levels and affecting their sensory development.”

In the abstract, again, we have replaced “disrupt” by “affect” (lines 47-49):

“Our results highlight that two different anthropogenic stressors can affect critical developmental and ecological transitions via the same physiological pathway.”

We have then replaced “sensory impairment” by “impacts on sensory development” in the final paragraph of the Introduction (lines 94-98):

“Here, we use the coral reef-dwelling convict surgeonfish, *Acanthurus triostegus*, to investigate the importance of the TH endocrine pathway on sensory development during metamorphosis, whether exposure to two distinct anthropogenic stressors (increased temperature and the waterborne organophosphate pesticide chlorpyrifos) can cause TH signaling disruption, and whether the resulting impacts on sensory development increase vulnerability to predation.”

Again, we have replaced “impair” by “affect” in the title of one subsection of our Result section (line 135):

“Increased temperature and chlorpyrifos affect TH levels and sensory development”

We have also replaced “impaired” by “affected”, “maturation” by “development”, “sensory organs” by “sensory structures” and “TH disruption” by “TH levels” in the following sentence of the result section (lines 152-154):

“Exposures to +3°C or CPF30 were therefore associated with comparably affected TH levels and development of sensory structures, but CPF30 fish experienced a lower T3/T4 ratio than +3.0°C fish (Supplementary Fig. 2).”

Then, we have also replaced “disrupt TH signaling” by “decrease TH levels” in another Results subsection title (line 165):

“Increased temperatures and chlorpyrifos synergistically decrease TH levels”

Similarly, we have replaced “maturation” by “development”, removed the term “disruptive”, replaced “TH disruption” by “decreased TH levels”, and replaced “developmental impairments” by “affected sensory development” in the first paragraph of our Discussion section (line 175-178):

“Our results highlight the key role an endocrine process (i.e. TH signaling) plays in regulating sensory system development in recruiting fishes, the effect anthropogenic stressors can have on this endocrine function, and the consequences of decreased TH levels and affected sensory development for determining the outcome of predator-prey interactions.”

In this same paragraph, we have also replaced “impaired sensory organ maturation” by “diminished sensory development” (lines 178-180):

“Fish with pharmacologically disrupted TH signaling experienced diminished sensory development and anti-predator behaviors, and were more vulnerable to predation.”

Again, in this paragraph, we have replaced “disrupted” by “affected” (lines 180-182):

“Sensory development and vulnerability to predation were comparably affected by two distinct anthropogenic stressors (increased temperature and CPF).”

Later in the discussion, we have replaced “impaired” by “affected” (lines 215-216):

“In addition to causing developmental defects, TH signaling disruption also affected the behavior and survivorship of metamorphosing fish.”

In the discussion, we have also replaced “causing TH disruption” by “decreasing TH levels” (lines 236-239):

“This variety of action modes may explain the synergistic effect of both stressors on TH levels with co-exposure to 1.5°C or CPF5 decreasing TH levels (Fig. 4a-b), while these stressor levels did not affect TH levels when exposed separately (Fig. 3a,b).”

According to the reviewer’s comment, we have also replaced “maturation” by “development” on 26 occurrences throughout the whole manuscript.

Similarly, we have complied with the reviewer’s comment by replacing “sensory organs” by “sensory structures” (6 occurrences throughout the entire manuscript), to encompass both the sensory organs (retina and lamellae) and the sensory accessory structures (trunk canal pores).

Overall, the terms “undermine” and “maturation” are no longer used in the manuscript, and the terms “impair/impairments” and “disrupt/disruptions” are only used when referring to results from other studies or when referring to broader mechanisms/hypotheses, but no longer when describing the precise results of this study.

Comment 2: That being said, behavioral responses to a visual predator stimulus, and to a chemical predator stimulus (“chemical” not “olfactory”; see caption in Supplemental Figure ___) were indeed affected by some aspects of experimental treatments, implying that changes in the timing of sensory development did indeed affect behavior. In addition, overall survival (# of fish surviving; not “survival rate” [line ____]) was affected by experimental treatments. However, in this case, behavior was not specifically observed – although overall behavioral alteration must have resulted in a change in the ability of predators to capture prey [and/or for

prey to avoid predators]). Again, the language used to describe the results of the experiments presented need to be modified so that they are not overstated; such changes in word usage will not change the outcomes of the experiments or the conclusions reached, but will make the report of the outcomes more precise.

R: We have removed the terms “rate/rates” as we agree with the reviewer that we are referring to the number of fish surviving. Similarly, we thank the reviewer for pointing the “olfactory” term that we had forgot to replace by “chemical” in the supplementary figure. The change has now been made.

Regarding the reviewer’s comment on the potential overstated results of our survival experiments, we would like to precise that we actually only refer to “anti-predator behaviors” in the manuscript when describing/discussing the results of the visual and chemical choice experiments, not the survival experiment. This is particularly evidenced on lines 178-180, as this sentence summarizes the sensory development results, the behavioral results, and the survival results:

“Fish with pharmacologically disrupted TH signaling experienced diminished sensory development and anti-predator behaviors, and were more vulnerable to predation”

Similarly, lines 216-217 concerns the visual and chemical preference results, not the survival experiments:

“Fish that experienced pharmacological TH disruption presented anti-predator behaviors that were comparable to pre-metamorphosed larvae”

In contrast, when referring to increased temperatures and chlorpyrifos treatments (for which we did not assess the visual and chemical preferences), we have been very cautious not to mention behavioral alterations as we did not test it (lines 229-230):

“exposure to high levels of increased temperature and CPF both inhibit sensory development and reduce a fish’s likelihood of avoiding predation”

To further emphasize this, we have brought supplemental precisions in the first paragraph of our discussion (lines 180-182):

“Sensory development and vulnerability to predation were comparably affected by two distinct anthropogenic stressors (increased temperature and CPF).”

Comment 3: One issue with behavioral experiment design and data analysis – fish that did not show a preference are important – why were they deleted from the analysis (line 569)?

R: We agree that fish that did not show a preference are important, and those fish were included in the analysis, as stated in our Methods (lines 549-552):

“Fish that did not make a clear choice between the two water sources but spent more than 50% of the time in the choice area were included in the analysis. This was done as we wanted to assess fish preference as well as the absence of preference.”

Actually, in both the visual and chemical choice experiments, the only fish that were removed from the analyses are the fish that remain immobile (to prevent side bias unrelated to a clear choice made by the fish) or which spent > 50% of the time in the “no choice area” (as our goal was to compare the time spent in the choice area 1 vs the time spent in the choice area 2).

Comment 4: It should also be mentioned that the auditory system may be affected by the

stressors.....It has been shown that larval fish respond to sound, so do not neglect this sensory system, which was not studied here.

R: Agreed. In addition to the reference to the effects of sound pollution on the ecology of fish recruitment already made in the introduction, we now also refer to this system in our discussion section (lines 210-213):

“These results highlight the sensitivity and complexity of the mechanisms underlying the actions of TH on sensory system development, offering an avenue for future research, in particular on the auditory system, as hearing was not studied here but is also used by larval fish during recruitment and impacted by stressors²³.”

Comment 5: Line 36/41 – please differentiate between metamorphosis and recruitment – these are two transitions, but “transition” (singular) is used in line 36.

R: Agreed. We have changed this sentence (lines 38-39):

“Larval metamorphosis and recruitment represent critical life-history transitions for most teleost fishes.”

Comment 6: Line 45/6 – suggestion – “Our results highlight the fact that two different anthropogenic stressors (physical, chemical) can alter critical developmental (morphological) and ecological transitions via the same hormonal pathway”

R: We thank the reviewer for this suggestion. We have modified this sentence (lines 47-49):

“Our results highlight that two different anthropogenic stressors can affect critical developmental and ecological transitions via the same physiological pathway.”

Comment 7: Line 60 – do you mean “metamorphosis” here?

R: Yes, we do mean “metamorphosis” here, as an example of a critical life-history transitions evoked in the previous part of the sentence.

Comment 8: Line 75-76 – I do not understand the logic in this sentence. Please check.

R: We confirm that this sentence is correct, and that no issue was raised by the previous 3 reviewers. This sentence indicates that anthropogenic stressors are known to affect predator and prey differently (i.e. affect fish prey more than fish predators), therefore having the potential to drastically reshape fish communities.

Comment 9: Line 85-87 – please delete “anthropogenic” – this is understood.

R: Agreed and changed.

Comment 10: Line 87 – here and throughout – do not use TH because in different parts of the MS you are talking about one form and in other places you are talking about two forms. Just spell it out.

R: TH is used when referring to both T₃ and T₄, and when referring to TH signaling more generally. When we only refer to either T₃ or T₄, we specifically used T₃ and T₄, and not TH. Therefore, we believe that our use of TH, here and throughout, is appropriate.

Comment 11: Line 106 – “rapid” – compared to what?

R: The term “rapid” refers to the development of new lamellae, a 50% increase in bipolar cell (bpc) density in the retina, and a 240% surge in trunk canal pore density, from d0 to d8 (i.e. in less than 8 days), as indicated in the sentence lines 104-107.

Comment 12: Line 108 – “surge” – compared to what? And is it # or density that was measured? Further, the # of pores is initially determined by the number of neuromasts in the canal (one pore on either side of a neuromast). So, is there a maximum number of pores reached during development?

R: The term “surge” refers to a 240% increase in trunk canal pore density. We believe that a 240% increase in less than a 8 days is a surge, and this term was not raised as an issue by the previous three reviewers. As indicated in our Methods section and supplementary materials, the length of the fully formed trunk canal (measured from the vicinity of the head to the base of the tail) does not change in *A. triostegus* during metamorphosis (Supplementary Fig. 9). The measure of the number of trunk canal pores is therefore equivalent to the measure of density of pores in the trunk canal, as the length of the trunk canal does not vary during metamorphosis. We do not know if there is a maximum number of pores reached during development, and according

to this comment and the comment 1, that is why we no longer use the term maturation, but rather the term “development”.

Comment 13: Line 112 – “increased rate of maturation” – does a statistical test support this? And please see my comments about “maturation” above.

R: Yes, all the statistic tests supporting this are presented in Fig. 1e-g. We agree with the reviewer’s comment about maturation and have made the appropriate changes (see our replies to comment 1 and 12).

Comment 14: Line 112/113 – pores are not sensory organs – see my comment about this above.

R: Agreed. To prevent any miswording here we have replaced throughout the whole manuscript “sensory organs” by “sensory structures” (see our reply to comment 1), to encompass both the sensory organs (i.e. lamellae and retina) and other sensory related structures (i.e. lateral line trunk canal pores).

Comment 15: Line 116 – did supplemental T3 rescue the retina or LL pores? If not, why not?

R: This was not tested and was previously discussed during the first revision of the manuscript. Briefly, for sampling limitations reasons, we only evaluated the effect of supplemental T₃ rescue for N3-treated fish on lamellae.

Comment 16: Line 122 – instead of CT in the text, can you spell out “control”? This would make the text easier to read.

R: Agreed and changed.

Comment 17: Line 130-131 – “survival” – does this refer to the number of fish that survived, or the total % of fish that survived? Please clarify.

R: According to this comment (but see also our reply to comment 2), we have added the following clarification (lines 128-130):

“We found that d2 T3-treated fish exhibited significantly higher survival (i.e. number of fish that survived) than d2 control fish, while d2 N3-treated fish exhibited lower survival than d2 control fish (Fig. 2c)”

Comment 18: Line 143 – bpc – would it be possible to spell this out, or put in caps throughout, and in graphs? Every time I see it, it looks like a typo to me.

R: Agreed. We have replaced “bpc” by “bipolar cell” throughout the entire manuscript, figures, and supplementary materials.

Comment 19: Line 154 – is it survival “rate” or the % of fish that survived? Check usage of “rate” throughout.

R: See our reply to comments 2 and 17. We no longer use the word “rate” when referring to these predation experiments.

Comment 20: Line 164 – change “co-exposed” to “simultaneously exposed”

R: Agreed and changed.

Comment 21: Line 165 – “suboptimal” – how was this determined?

R: This was determined with the results obtained with each single stressor, separately, for which +1.5°C, CPF1, and CPF5, had no significant effect of TH levels. This term is classically used in endocrinology. For better clarity, we have added this precision (lines 166-168):

“To test whether exposure to both stressors had additive or interacting effects, we simultaneously exposed d0 larvae to different suboptimal dose regimes of both stressors (i.e. dose regimes that were found to individually have no significant effect on TH levels) until d2.”

Comment 22: Line 172 – “recruiting fishes” – perhaps say “settlement stage fishes”?

R: This comment is contradictory with the comment made by the reviewer 1 of the first round of revision, during which we were asked to clarify the use of the terms “recruitment”, “settlement”, and “metamorphosis”. Following this, we believe that the use of “recruiting fishes” is here more appropriate, and clearer to the readers.

Comment 23: Line 176 – it is “presumed” anti-predator behaviors – behavior was not measured.

R: As mentioned in our reply to comment 2, behavior was measured in fish for which we manipulated TH signaling pharmacologically (Fig. 2a-b). The sentence mentioned by the reviewer clearly states (lines 178-180): “Fish with pharmacologically disrupted TH signaling experienced diminished sensory development and anti-predator behaviors”. We therefore do not feel that there is a need to make the change asked by the reviewer here.

Comment 24: Line 178 – “apparently” recovered their ability to avoid.....

R: Agreed and changed.

Comment 25: Line 193 – “Promoted TH signaling”, should be “enhanced TH signaling...”

R: Agreed and changed.

Comment 26: Line 196 – “rescued maturation _____ of their olfactory organ....” (but again, consider whether “maturation” should be used).

R: Agreed, and see our replies to comments 1 and 13 where we explain that, following the reviewer’s recommendation, we no longer use the term “maturation”.

Comment 27: Line 199 – development (not maturation) of the trunk canal does include differentiation of canal neuromasts (not studied here) and enclosure and elaboration of the canal. The epithelial proliferation that occurs, that is relevant here, is the formation and elongation of the pores in the skin covering the lateral line canal.

R: Agreed and changed (lines 204-207):

“In contrast, the development of the trunk canal involves both the differentiation of canal neuromasts (i.e. neurogenesis, not studied here) and the enclosure and elaboration of the canal itself⁴³. The epithelial proliferation that occurs, that is relevant here, is the formation and elongation of the pores in the skin covering the lateral line canal.”

Comment 28: Line 211 – “increased mortality” – OK, but refer to “higher numbers of fish captured by predators”, which is presumably due to a change in the developmental timing of the sensory organs.

R: We agree with the reviewer and this is the reason why we stated “increased mortality during a predation experiment”, which to us means “increased mortality (i.e. higher numbers of fish captured by predators)”.

Comment 29: Line 213 – 217 – please break into two sentences.

R: Agreed and changed (lines 220-223):

“We found that T₃-treated fish exhibited fastened sensory development (Fig. 1e-f) and comparable responses to predator cues to control fish (Fig. 2a-b), which is consistent with an accelerated sensory development facilitating more effective anti-predator abilities. Nevertheless, we cannot rule out the possibility that supplemental T₃ may have also impacted their behavior in other ways.”

Comment 30: Line 261 – delete “detrimental” and add “changes behavior” at the end of the sentence.

R: We have added “behavior”. While we agree to replace “impair / impairments / undermine / disrupt / disruption” by “affect” (see our reply to comment 1), our results demonstrate how stressors affect sensory development and behavior with severe impact on fish survival facing predation. To us, this is a detrimental consequence and we therefore believe that this should be indicated. We have therefore replaced “detrimental” by “adverse”, as this lower down the tone of this sentence while still indicated the “trend” of the consequences of the exposure to stressors.

Comment 31: Line 265 – say “populations and communities”.

R: Agreed and changed.

Comment 32: Line 285 – say “metamorphosis is quickly followed by settlement and recruitment” (not “coincides”) – this is a good place to reinforce that these are three different processes.

R: We agree that these are different processes, and while we acknowledge that metamorphosis may be followed by settlement and recruitment in some species, this is not the case in *Acanthurus triostegus* and many other coral reef fish species. Indeed, metamorphosis and recruitment coincide in these species (see Holzer et al. 2017 – eLife 6:e27595). We therefore prefer to leave this sentence as it is.

Comment 33: Line 362 – This sentence should be in the Intro to the paper, where the relevance of the stressors are laid out.

R: Our introduction puts a great emphasis on the effects of stressors on fish recruitment processes, but not specifically on each specific stressor. The relevance of each specific stressor is rather discussed in our Discussion section, where we have therefore moved this sentence (lines 247-249):

“Organophosphates are the most widely used pesticide globally, and CPF is the second most commonly detected pesticide in water, frequently occurring in temperate and tropical coastal runoff⁶⁴”

Comment 34: Line 409 – say “Retina”

R: Agreed and changed.

Comment 35: Line 427 – please spell out “ds”

R: Agreed and changed.

Comment 36: Line 429 – should be “another acanthurid species”

R: Agreed and changed.

Comment 37: Line 432 – “processing efficiency of the retina toward visual cues” – please clarify. Perhaps “the ability of the retina to form an image”?

We agree that this was not sufficiently clear. Given that the function of bipolar cells is already clearly explained previously in this paragraph (lines 424-425), we have removed this unclear statement. The sentence now reads (lines 435-437):

“We prioritized the examination of bipolar cell densities as important changes in bipolar cell density was observed during *U. tragula* metamorphosis¹³.”

Comment 38: Line 439 – do you mean “thickness” instead of “width”?

R: Yes, thank you for advising this change. We have replaced “width” by “thickness”.

Comment 39: Line 448 – say “number of folds or lamellae” – to define lamellae.

R: Agreed and changed.

Comment 40: Line 450 – change “circulating” to “moving through each....”

R: Agreed and changed.

Comment 41: Line 454 – shorten to “2.5% glutaraldehyde in 1M sucrose and 0.1M sodium cacodylate (pH 7.4)...”. Make same changes in line 456-7.

R: Agreed and changed.

Comment 42: Line 476 – should be “the lateral line system and are ciliated sensory organs, composed of hair cells, like those in the inner ear, located....”

R: Agreed and changed.

Comment 43: Line 479 – “at the level of the canal pore” is incorrect – not sure what is being described here. The neuromasts are within the short canal segments between pores.

R: We thank the reviewer for this clarification. We have reworded our sentence (lines 480-482):

“Canal neuromasts are found in the epithelium lining the bottom of the lateral line canals, and one canal neuromast is usually found within the short canal segments between two adjacent canal pores⁴³.”

Comment 44: Line 486 – what are “vertical plates” – are these the canal walls such that the roof of the canal is still just soft tissue?

R: Vertical plates are structures present on *Acanthurus triostegus* body at the metamorphic stage. They are not scales, nor canal walls. Their definition is well explained and imaged by the reference Frédérich et al., 2010 – Journal of Applied Ichthyology, which we cite. The goal of this sentence is to say that *A. triostegus* does not exhibit scales at this stage, and that the trunk canal is only composed of soft tissue. We understand from the reviewer’s comment that the reference to vertical plates was therefore misleading, and we have therefore clarified this sentence, which now reads (lines 488-489):

“At this stage, *A. triostegus* does not exhibit scales⁶⁹, and the trunk canal is therefore only composed of soft tissue (Supplementary Fig. 4a).”

Comment 45: Line 494 – “numbers” – yes, but please check throughout where “density” may have been used.

R: It is because there is no variation in the length of the trunk canal during metamorphosis that a variation in the number of trunk canal pores reflects a variation in the density trunk canal pores (i.e. an increased number of pores corresponds to an increased density of pores).

Comment 46: Line 519 – “odor” suggests olfaction. Can you just say “chemical cues from predator”?

R: Thank you for pointing this reference to olfaction that we forgot to remove. We have modified this text. The text now reads (lines 519-523):

“Each trial presented an individual fish with two water sources: control seawater (\emptyset = UV-sterilized and 1 μm filtered seawater from the collection site) vs ‘seawater containing chemical cues from predator’ (P = UV-sterilized and 1 μm filtered seawater from the collection site in which five *Lutjanus fulvus* predators were soaked, into a 125 L tank, for two hours prior to the experiment).”

Comment 47: Line 530 – arena or area – please check throughout and in figures and figure captions (main and supplementary).

R: We apologize for these inconsistencies. The proper term is “area” when referring to choice tanks, and “arena” when referring to the predation arena. We have fixed this throughout the text, figures, and figure captions (in both the main text and the supplementary materials).

Comment 48: Line 533 – Not sure what immobility has to do with anything about side preference. Please clarify and check throughout.

R: We agree with the reviewer that immobility does not inform about side preference. An immobile fish rather reflects a fish that is stressed or that has not explored the different stimuli presented in the choice aquaria. That is the reason why immobile fish were removed from the

analysis, as clearly indicated in our manuscript. See also our reply to comment 3 for further explanation.

Comment 49: Line 537 – was this “in the dark with IR light” or was it “in the dark with red light”?

R: We mean “in the dark with red light” and have now clarified this in the text (line 539):

“These experiments were performed in the dark with red light”

Comment 50: Line 552 – should be “in the presence of a visual predator”.

R: Agreed and changed.

Comment 51: Line 594 – should be “and the Fulton’s K condition factor was used for pharm.....treatments.”

R: Thank you for pointing this typo. We have reworded this sentence, which now reads (lines 594-596):

“Linear models (LM) were used to assess if pharmacological treatments influenced the bipolar cell, ganglion cell and photoreceptor nuclei densities, and if pharmacological and anthropogenic stressor treatments influenced fish Fulton’s K condition factor.”

Comment 52: Line 595 – varied with age, not varied across metamorphosis; was d8 post-metamorphic?

R: According to Holzer et al., 2017 - eLife 6:e27595, d8 can be considered as post-metamorphic. We have replaced “across metamorphosis” by “with age”, as recommended by the reviewer.

Comment 53: Line 599 – should be “due to the fact that the analysis was carried out by two different people in two different years”.

R: We apologize if we have not been clear enough here. The reason is that different Cobas analysers (i.e. machines, not humans) were used in the two different years, as indicated in our text, and not that different people carried out the experiments in two different years. For better clarity, we have reworded the sentence, which now reads (lines 599-601):

“TH level or T₃/T₄ ratios were used as the dependent variable, and replicate was included as a random factor to account for differences in TH levels only due to the two different Cobas analysers that were used in the two different years.”

Comment 54: In ALL Graphs – please add the units of the variables on the Y-axis (e.g., # of lamellae, bpc density (#cells/____), # trunk canal pores), etc.

R: Thank you for pointing these inconsistencies. We have now added the units of the variables in all graphs, when this was needed.

*Comment 55: In all captions, results of statistical tests with P-values and F values must be added, esp. since these are not in the text. Then they can be eliminated from the figures themselves (use ** instead), which will make them less cluttered.*

R: Given that there are up to 12 statistical tests presented per figure, we believe that the results of statistical tests are better conveyed when presented on the figure directly. Please note that this has not been raised as an issue by any of the three previous reviewers, nor by the editor, and that actually the clarity of our figures has even been acknowledged during the previous round of reviews. We therefore disagree that results of statistical tests should be presented in figure captions, where they would be lost in endless lists of equations, which would make them challenging for readers to read and to relate to figure panels and subpanels.

Comment 56: Figure 1 is quite information rich – Can the image be split from the graphs with the addition of the image in Supplementary Fig. 4a – for anatomical context. Similarly – can the image showing the nares in the Supplementary Figure be added here to accompany the rosette for context? These are likely not commonly understood since they are “fish-specific”. However, the cross section of the retina should be generally recognizable.

R: We believe that the current presentation allows readers to better understand which graph (Fig. 1e-g) is related to which sensory structure (Fig. 1b-d). Dividing this figure in two figures would actually break this link and make the interpretation of the results more challenging to the readers. Concerning the supplementary figures, since *Nature Communications* is open access, both the manuscript and the supplementary materials will be available to all readers if our manuscript gets

accepted for publication. It is very important to us to maintain a clear link between each sensory structure anatomy and their respective development during metamorphosis. We therefore prefer not to divide figure 1 into two separate figures, and to maintain the supplementary anatomical information about each sensory structure in the supplementary materials.

Comment 57: Suppl. Fig. 3 – “rosetta” should be “rosette” and add text to say how image b is related to image a (what was dissected away?). “SEM picture” should be “scanning electron micrograph”

Suppl. Fig. 4 – Fig. a should be moved to Fig. 1 – see comment above, then this figure can be eliminated, since b is in Fig. 1.

R: Thank you for these corrections that clarify our figure caption, which now reads (lines 21-27 – Supplementary Materials):

“Supplementary Fig. 3. Olfactory organ location and morphology in a d0 *Acanthurus triostegus*.

a Scanning electron microscope (SEM) photograph of *A. triostegus* head (right side of the head) localized around the right olfactory organ (white dotted-rectangle) between the eye and the snout. Dotted area indicates the region that is dissected (removal of the skin layer between the two nares) and presented in b. Scale bar indicates 1 mm. b SEM photograph of a rosette with 11 lamellae (light green dots). Scale bar indicates 100 μm .”

Please see our reply to comment 56 concerning the move of figure panels.

Comment 58: Supple Fig. 5 – change “ganglionar” to “ganglionic”, “layer width” should be “layer thickness” – see other comment above and check throughout.

R: Agreed and changed.

Comment 59: Suppl. Fig 10 – should be “chemical” not “olfactory” in caption and figure.

R: Agreed and changed.

Comment 60: Suppl. Fig. 12 – please add scale bar or mention tank size in caption.

R: We have added this information (line 92 – Supplementary Materials):

“Arena size: 1 x 1 x 1 m (L x W x H)”

Response to Reviewer 2:

General comment: *All concerns have been adequately addressed.*

R: We would like to thank Reviewer 2 for this second review of our manuscript.